# IFN-γ-dependent NK cell activation is essential to metastasis suppression by engineered *Salmonella*

Qiubin Lin[1,2,7], Li Rong [1,7], Xian Jia [3], Renhao Li[1,4], Bin Yu[1], Jingchu Hu [5], Xiao Luo [5], S. R. Badea [1], Chen Xu[1], Guofeng Fu[3], Kejiong Lai[3], Ming-chun Lee[1], Baozhong Zhang[5], Huarui Gong [1], Nan Zhou [5], Xiao Lei Chen [3,6], Shu-hai Lin [3✉], Guo Fu [3,6✉] & Jian-Dong Huang [1,2,5✉]

Metastasis accounts for 90% of cancer-related deaths and, currently, there are no effective clinical therapies to block the metastatic cascade. A need to develop novel therapies specifically targeting fundamental metastasis processes remains urgent. Here, we demonstrate that *Salmonella* YB1, an engineered oxygen-sensitive strain, potently inhibits metastasis of a broad range of cancers. This process requires both IFN-γ and NK cells, as the absence of IFN-γ greatly reduces, whilst depletion of NK cells in vivo completely abolishes, the anti-metastatic ability of *Salmonella*. Mechanistically, we find that IFN-γ is mainly produced by NK cells during early *Salmonella* infection, and in turn, IFN-γ promotes the accumulation, activation, and cytotoxicity of NK cells, which kill the metastatic cancer cells thus achieving an anti-metastatic effect. Our findings highlight the significance of a self-regulatory feedback loop of NK cells in inhibiting metastasis, pointing a possible approach to develop anti-metastatic therapies by harnessing the power of NK cells.

[1] School of Biomedical Sciences, Li Ka Shing Faculty of Medicine, The University of Hong Kong, Pokfulam, Hong Kong SAR, China. [2] HKU-Zhejiang Institute of Research and Innovation (HKU-ZIRI), Hangzhou, China. [3] State Key Laboratory of Cellular Stress Biology, Innovation Center for Cell Signaling Network, School of Medicine, Xiamen University, Xiamen, China. [4] Department of Medicine, Li Ka Shing Faculty of Medicine, The University of Hong Kong, Pokfulam, Hong Kong SAR, China. [5] Institute of Synthetic Biology, Shenzhen Institutes of Advanced Technology, Chinese Academy of Sciences, Shenzhen, China. [6] Cancer Research Center of Xiamen University, Xiamen, China. [7] These authors contributed equally: Qiubin Lin, Li Rong. ✉email: shuhai@xmu.edu.cn; guofu@xmu.edu.cn; jdhuang@hku.hk

Metastasis accounts for 90% of cancer-related deaths and the blocking of metastatic cascade has critical clinical impacts[1]. Metastasis of cancer cells to distal organs is a complex multistep process in which cancer cells migrate from the primary tumor, enter into the circulation (intravasation), and then exit the circulation (extravasation) at distant organs, leading to metastatic colonization[2]. Metastatic tumor cells dynamically interact with the microenvironment and most of them are likely to die, especially during seeding and colonization at distant organs. Few survived metastatic tumor cells finally form metastatic tumors that are markedly different from the primary tumor, which leads to the original effective cancer therapy in the primary tumor having limited or no therapeutic effect on metastatic tumors[3,4]. However, current preclinical and clinical development of cancer therapies, including cancer immunotherapies, is initially evaluated, largely depending on their ability to suppress tumorigenesis and/or primary growth rather than their anti-metastatic activity[5]. Therefore, there is an urgent need for novel therapeutic strategies and agents targeting fundamental metastatic processes. Previously, we demonstrated that treatment with an engineered *Salmonella typhimurium* strain YB1[6] showing reduced toxicity against the host could not only inhibit orthotopic liver tumor growth but also lung metastasis[7]. Similar phenomena have previously been reported occasionally in some other *Salmonella* strains[8–10]. However, it is unknown whether this is a general anti-metastatic effect and the underlying mechanisms remain unaddressed.

*S. typhimurium* is a facultative anaerobic pathogen that can colonize tumors. Besides its use as a delivery system for antitumor therapeutic agents, it also possesses an intrinsic anti-tumor effect, largely attributed to its immunomodulatory activity[11]. Systemic administration of *S. typhimurium* can effectively stimulate the immune system, resulting in the increased production of systemic proinflammatory cytokines, such as interleukin (IL)-1β, IL-18, tumor necrosis factor-α (TNF-α), and interferon-γ (IFN-γ), as well as activation of both innate and adaptive immune cells[11,12]. These manipulated immune responses might lead to a hostile environment for tumor progression. For example, *S. typhimurium* treatment in mice bearing CT26 tumors was reported to suppress the growth of primary tumor through increased production of TNF-α and IL-1β by macrophages and dendritic cells[13]. Likewise, *S. typhimurium* treatments in other different contexts were reported to promote the recruitment of neutrophils, granulocytes, and macrophages, as well as activation of CD8+ T cells and natural killer (NK) cells[14,15]. However, the roles of these *Salmonella*-induced immune responses in metastasis suppression remains unclear.

In the present study, we show that engineered *Salmonella* effectively suppresses metastasis of a broad range of cancers and this process only requires innate immune responses. Among the many induced cytokines, we identify IFN-γ as an indispensable factor for inhibiting lung metastasis. Based on CyTOF (mass cytometry or cytometry by time of flight) analysis of the immune responses after *Salmonella* treatment and antibody-mediated cell depletion, we further demonstrate that NK cells are the major cell population involved in *Salmonella*-provoked metastasis suppression. We find that NK cells secrete IFN-γ, which in turn promotes the accumulation, activation, and cytotoxicity of NK cells, generating a self-sustaining feedback loop. IFN-γ and NK cells are indispensable for *Salmonella* to suppress cancer metastasis.

## Results

### Engineered *Salmonella* inhibits cancer metastasis in multiple syngeneic mouse tumor models.

We observed that infection with *Salmonella* YB1, an engineered oxygen-sensitive strain based on the wild-type *S. typhimurium* SL7207[6], had similar inhibitory effects on lung metastasis in two different metastasis models established with murine mammary carcinoma 4T1 cell line in BALB/c mice (Fig. 1a–e). The orthotopic metastasis model treated with YB1 exhibited only a slight delay in the primary tumor growth, but lung metastasis was significantly reduced (Fig. 1a, b). When the primary tumors were surgically removed 1 week after treatment, 44% of mice treated with YB1 survived metastasis-free for more than 60 days, whereas all the mice in the control group died of lung metastasis within 26 days (Fig. 1c and Supplementary Fig. 1a). The experimental metastasis model, in which mice were pretreated with YB1 and then inoculated with cancer cells intravenously (i.v.) through the tail vein to establish lung metastasis (Fig. 1d), showed the YB1 treatment completely inhibited the formation of metastasis in the lung (Fig. 1e). Notably, the anti-metastatic activity provoked by a single dose of YB1 was able to last for at least 2 weeks (Supplementary Fig. 1b, c).

To determine whether the inhibition of lung metastasis by YB1 was cancer type-dependent or host genetic background-dependent, we extended the analyses to several other experimental metastasis models based on different cancer cell lines and mouse strains. In addition to the murine mammary carcinoma 4T1 cell line with epithelial morphology from a BALB/c background[16], we used a murine colon carcinoma CT26 cell line with fibroblast morphology from a BALB/c background[17] to represent different cancer types from the same background, which showed similar inhibitory effects on lung metastasis upon YB1 treatment (Fig. 1f). Besides cell lines derived from solid tumors, we also tested a mouse lymphocytic leukemia L1210 cell line, which metastasizes to the bone marrow, blood, and other organs[18,19]. Treatment with YB1 6 days after i.v. inoculation of L1210 cells effectively suppressed cancer cell homing to the bone marrow (Supplementary Fig. 1d–f). We extended the metastasis model from BALB/c to C57BL/6 mouse strain, to represent a different mouse genetic background. A bladder carcinoma MB49 cell line and a melanoma B16F10 cell line from C57BL/6 mouse background[20] were used to establish experimental metastasis models in C57BL/6 mice, which also showed almost complete inhibition of lung metastasis upon YB1 treatment (Fig. 1g, h). In general, we found the anti-metastatic activity induced by YB1 treatment was dependent on the dose of YB1 (Fig. 1i). Although the parent *Salmonella* SL7207 strain showed similar anti-metastatic effects as *Salmonella* YB1 (Supplementary Fig. 1g), YB1 is much less toxic against the host and causes almost no side effects[6,21] (Supplementary Fig. 1h, i). Altogether, these findings suggest a possible general mechanism that the anti-metastatic activity of *Salmonella* is dose-dependent, but independent of cancer type and host genetic background.

### *Salmonella* treatment interferes with early metastatic cascade and inhibits early survival of cancer cells in the lung.

The complex process of metastasis includes localized invasion, intravasation, circulation, extravasation, and colonization[2]. Interference at any stage by *Salmonella* could lead to a reduction of lung metastasis. We, therefore, evaluated the role of *Salmonella* on the different stages of the metastatic cascade. The 4T1 orthotopic metastasis mouse model was treated with YB1 at different time points (7, 12, and 19 days) after implantation of 4T1 to the fat pad on day 0. As shown in Fig. 2a–f, treatment with YB1 on day 7 showed the best anti-metastatic effect, whereas treatment on day 19 failed to inhibit lung metastasis. To monitor metastatic status, mice with no YB1 treatments were killed at each time point to examine lung metastasis, which was only visible on day 19 (Fig. 2e), indicating YB1 could interfere at the early stage of metastasis, but not after metastasis established.

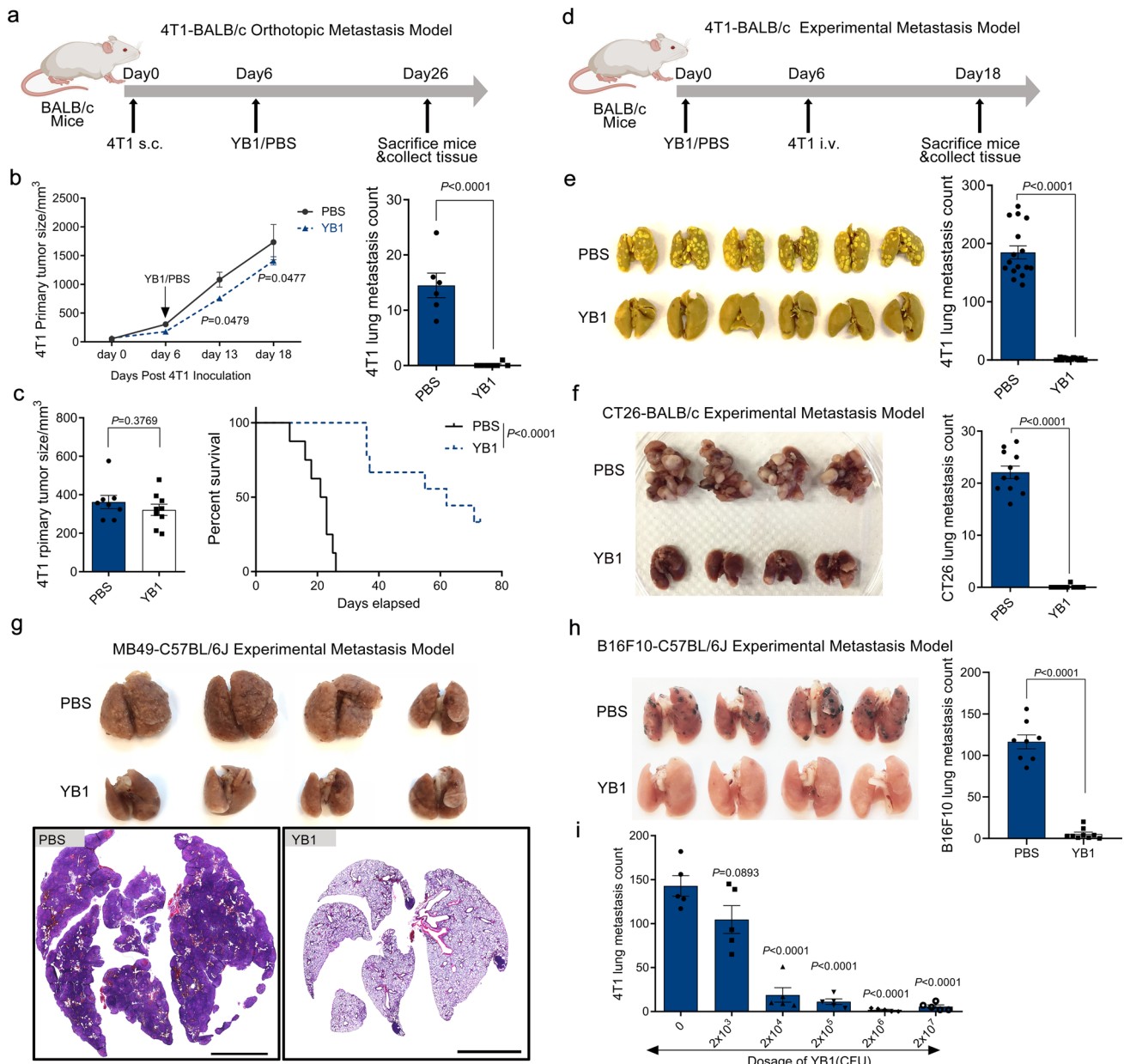

**Fig. 1 *Salmonella* YB1 treatment inhibits cancer metastasis in multiple syngeneic mouse tumor models. a** Overall procedures to establish 4T1-BALB/c orthotopic metastasis model. Cancer cell inoculation was followed by treatment with YB1 or PBS. **b** Quantification of 4T1 primary tumor size (two-sided multiple *t*-tests) and relevant lung metastasis (unpaired two-tailed *t*-test) after YB1 treatment in the 4T1-BALB/c orthotopic metastasis model ($n = 6$ PBS, $n = 7$ YB1). **c** Mice tumor size measured (unpaired two-tailed *t*-test) before surgical removal of 4T1 primary tumors and relevant Kaplan–Meier survival curve (log-rank test) of these mice ($n = 8$ PBS, $n = 9$ YB1). **d** Procedure to establish 4T1-BALB/c experimental metastasis model. 4T1 cells were i.v. injected into BALB/c mice pretreated with YB1 or PBS. **e** Picture of lungs fixed in Bouin solution and quantification (unpaired two-tailed *t*-test, displayed is a combined result of three independent experiments, $n = 16$ per group) of 4T1 lung metastases after YB1 treatment in the 4T1-BALB/c experimental metastasis model. **f** Picture of lungs and quantification (unpaired two-tailed *t*-test, $n = 11$ per group) of lung metastases after YB1 treatment in an experimental metastasis model established by colon cancer (CT26) in BALB/c mice. **g** Picture of 4% PFA-fixed lungs and representative H&E staining of lung tissue in an experimental metastasis model established by bladder cancer (MB49) in C57BL/6J mice ($n = 4$ mice per group). Scale bar, 3 mm. Condensed tumor nodules were stained with H&E from PBS samples. **h** Picture of lungs and quantification (unpaired two-tailed *t*-tests) of lung metastases after YB1 treatment in the experimental metastasis model established by inoculation of melanoma (B16F10) in C57BL/6J mice ($n = 8$ PBS, $n = 9$ YB1). **i** Quantification of 4T1 lung metastases in the 4T1-BALB/c experimental metastasis model after treatment with different doses of YB1 ($n = 5$ per group). *P*-values are comparisons between no YB1 treatment and different dosages of YB1 treatment using unpaired two-tailed *t*-tests. Graphs and images presented are one representative experiment of two independent experiments (**b**, **c**, **g**, **i**). Graph presented is a combined result of two independent experiments (**f**, **h**). All data are presented as mean values ± SEM. Source data are provided as a Source Data file.

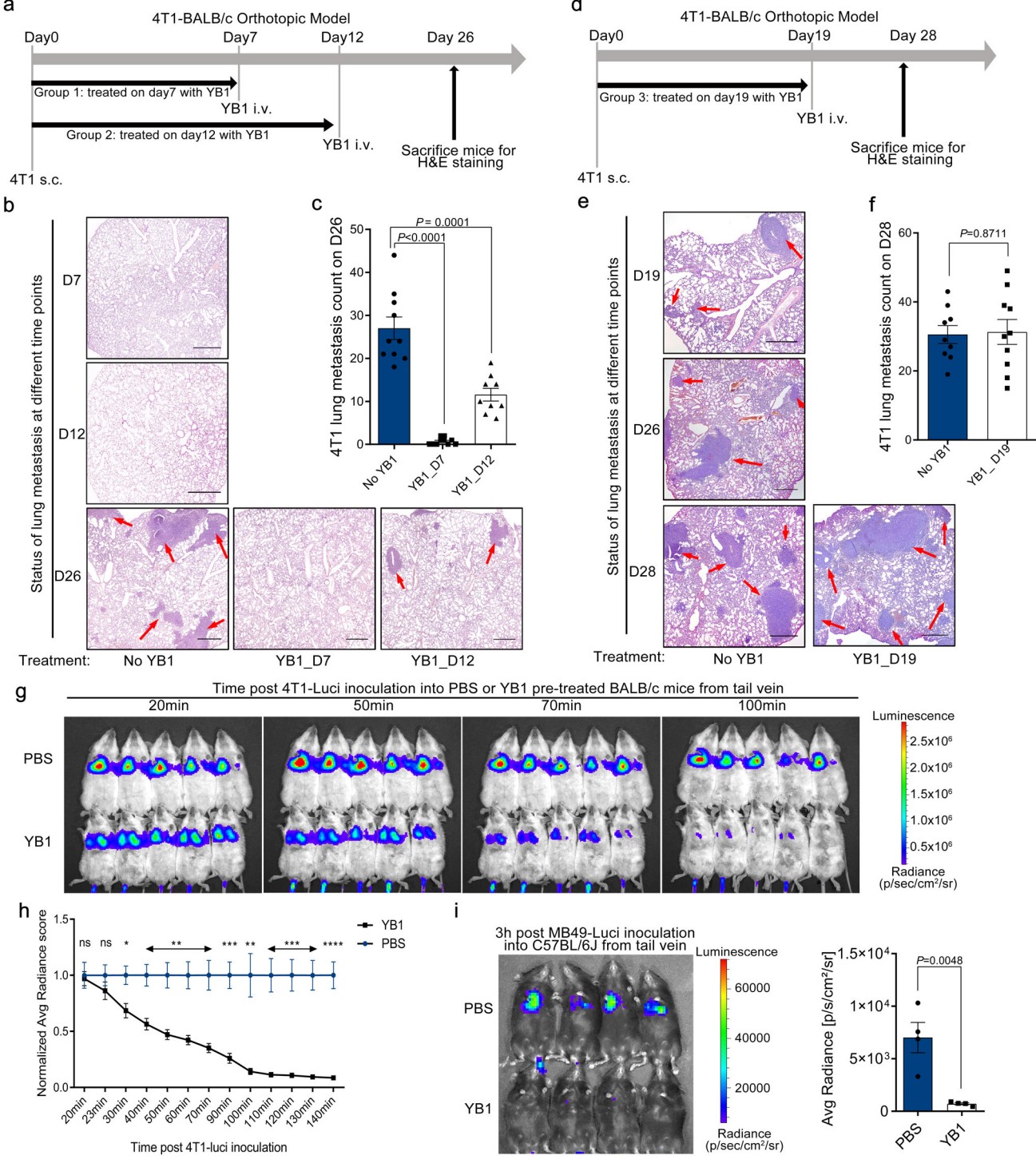

We hypothesized that the invasiveness of cancer cells from the primary tumor might be affected by YB1. Surprisingly, tumor sections stained for epithelial-mesenchymal transition (EMT) markers[22] showed elevated expression of vimentin and down-regulated E-cadherin at 54 h post infection, which was restored by day 7 (Supplementary Fig. 2), suggesting *Salmonella* YB1 treatment may even induce EMT within a short period of time, rather than inhibit EMT.

Even though a significant reduction of lung metastasis was observed, there were no indications that cancer cells became less invasive after YB1 treatment. Hence, we next examined the metastatic colonization steps. To track the metastatic colonization

process to the lung, we i.v. injected luciferase-labeled 4T1 cells (4T1-Luci) into BALB/c mice pretreated with YB1 or phosphate-buffered saline (PBS), which was then monitored by luciferase live imaging. Shortly after the injection, 4T1-Luci cancer cells accumulated in the lungs of both groups, but the luciferase signal quickly diminished in YB1-pretreated mice (Fig. 2g, h). A similar phenomenon was observed in the MB49-Luci-C57BL/6J experimental metastasis model. At 3 h post injection, MB49-Luci cancer cells accumulated in the lungs of control mice, whereas the luciferase signal again diminished in YB1-pretreated mice (Fig. 2i). Altogether, these data demonstrated that the colonization process of metastatic cascade was impaired by the YB1

**Fig. 2** *Salmonella* **YB1 treatment interferes with survival and early colonization of newly deposited cancer cells. a** On day 0, 4T1 cells were implanted into the fat pad of BALB/c mice. Two groups of mice were treated with YB1 on day 7 (YB1_D7, *n* = 10) or 12 (YB1_D12, *n* = 9), respectively. PBS as a control treatment (no YB1, *n* = 10). Lung metastasis was quantified on day 26. At each time point of the YB1 treatment, the status of lung metastasis was checked by H&E staining. **b** Representative H&E staining of lungs from mice killed on day 7 or 12 without YB1 treatment and those killed on day 26 with indicated treatments. **c** Quantification of 4T1 lung metastases (unpaired two-tailed *t*-tests) from mice as in **a**. **d** YB1 was performed on day 19 (YB1_D19, *n* = 10) after 4T1 implantation into BALB/c mice. PBS as a control treatment (No YB1, *n* = 9). Lung metastasis was quantified on day 28 (**f**). Extra mice were killed to check the status of lung metastasis by H&E staining on day 19 and 26 without YB1 treatment. **e** Representative H&E staining of lungs from indicated mice. **f** Quantification (unpaired two-tailed *t*-test) of 4T1 lung metastases on day 28. **g, h** Tracking and quantification of 4T1-Luci cells (two-sided multiple *t*-tests, *p* = 0.0163 (30 min), 0.0039 (40 min), 0.0024 (50 min), 0.0019 (60 min), 0.0015 (70 min), 0.0004 (90 min), 0.0023 (100 min), 0.0004 (110 min), 0.0002 (120 min), 0.0001 (130 min), and 0.00006 (140 min)) by luciferase live imaging after i.v. injection of 4T1-Luci into BALB/c mice pretreated with YB1 or PBS 3 days in advance (*n* = 5 per group). *P < 0.05, **P < 0.01, ***P < 0.001, ****P < 0.0001. **i** Tracking and quantification (two-tailed unpaired *t*-test) of MB49-Luci cells by luciferase live imaging 3 h after i.v. injection of MB49-Luci into C57BL/6J mice pretreated with either YB1 or PBS 3 days in advance (*n* = 4 per group). **b, e** Red arrows indicate tumor tissues. Scale bar, 300 μm. **c, f** Displayed is a combined result of two independent experiments. **g–i** Displayed is one representative experiment of two independent experiments. All data are presented as mean values ± SEM. Source data are provided as a Source Data file.

treatment, particularly for the early survival of cancer cells. Cancer cells were still able to accumulate at the lung but failed to establish metastasis after *Salmonella* treatment.

**IFN-γ is required for live *Salmonella* to suppress cancer metastasis.** In the experimental metastasis model, mice were inoculated with cancer cells after the *Salmonella* was cleared (Supplementary Fig. 3a), indicating a direct interaction between *Salmonella* and cancer cells was unlikely. We hypothesized that *Salmonella*-induced immune responses were responsible for blocking the metastatic cascade. In agreement with previous studies[14,23], *Salmonella* treatment induced systemic and tumor-localized proinflammatory immune responses. *Salmonella* treatment in the orthotopic metastasis model led to elevated levels of TNF-α, IL-6, and IL-1α in tumors (Supplementary Fig. 3b). Bacterial LPS (lipopolysaccharides) is a well-studied immunogen capable of inducing inflammatory responses and may act as a mediator to suppress metastasis by *Salmonella*. However, VNP20009, an attenuated msbB-deficient *S. typhimurium* strain with low LPS-associated toxicity[24], showed comparable anti-metastasis ability to *Salmonella* YB1 and treatment with LPS from *Salmonella* even promoted lung metastasis (Supplementary Fig. 3c). Moreover, compared with the control groups (PBS, heat-killed *Salmonella*, and non-pathogenic *Escherichia coli* strain DH10B), only live *Salmonella* YB1 cells were capable of effectively suppressing cancer cell metastasis and inducing strong cytokine responses such as IL-6, IL-12p70, IL-1β, TNF-α, IL-18, and IFN-γ (Fig. 3a, b). Suppression of inflammatory responses with a steroid drug prednisolone partially impaired the anti-metastatic activity of YB1 (Fig. 3c), confirming the hypothesis that *Salmonella*-induced inflammation plays an important role in the inhibition of cancer metastasis.

Next, we attempted to identify the immune factors necessary for suppressing metastasis after YB1 treatment. Among the increased systemic proinflammatory cytokines, TNF-α and IFN-γ were frequently reported to have anti-tumor activity. Thus, we hypothesized that they might play an important role in the anti-metastatic activity of YB1. We depleted TNF-α and IFN-γ in vivo using specific neutralizing antibodies. Interestingly, depletion of IFN-γ, but not TNF-α, completely abolished the anti-metastatic activity of YB1 (Fig. 3d). We next depleted IFN-γ in mouse metastasis models established by 4T1-Luci with live imaging. We found the depletion of IFN-γ dramatically diminished the inhibitory effect of YB1 on the metastatic colonization of cancer cells in the lungs (Fig. 3e). To confirm the role of IFN-γ in mediating the anti-metastatic activity of YB1, we established an experimental metastasis model by injecting MB49 cells into IFN-γ-knockout mice[25] and wild-type C57BL/6J mice. As expected,

YB1 inhibited cancer cell metastasis only in wild-type mice but not in IFN-γ-knockout mice, indicating that IFN-γ is a necessary cytokine for the YB1-induced suppression of metastasis (Fig. 3f and Supplementary Fig. 3d, e). However, i.v. administration of IFN-γ into wild-type mice failed to reduce lung metastasis in the experimental metastasis model (Fig. 3g), indicating that IFN-γ alone is not sufficient to inhibit metastasis. The localized IFN-γ responses induced by YB1 were tissue-dependent. On day 5 after YB1 treatment, we found IFN-γ levels were decreased in tumors but highly elevated in the lung (Supplementary Fig. 3f). Likely, the spatiotemporal kinetics of YB1-induced IFN-γ in vivo cannot be recapitulated by i.v. injection, besides tissue-specific levels of IFN-γ are yet not possible with current technologies. Altogether, these results showed that IFN-γ is one of the necessary factors for the anti-metastatic activity of *Salmonella* YB1.

**Innate immunity is sufficient for *Salmonella*-induced cancer metastasis suppression.** To explore which branch of the immune system is responsible for suppressing metastasis after YB1 treatment, we first investigated tumor-infiltrating T cells. Intriguingly, we found that levels of tumor-infiltrating T cells were dramatically decreased on days 1 and 5 after YB1 treatment in the 4T1 orthotopic metastasis model, whereas T-cell levels in lung tissue were not altered (Supplementary Fig. 4a–c). In addition, both CD4+ and CD8+ T cells were significantly reduced in the spleen (Supplementary Fig. 4d) and CD4+ T cells were reduced in tumor-draining lymph nodes (LNs; Supplementary Fig. 4e). We suspected that adaptive immunity might not be critical for YB1-induced inhibition of cancer cell metastasis. Therefore, we established both orthotopic and experimental metastasis models using immunodeficient NOD SCID (NOD.CB17-Prkdc^scid/J) mice, in which no functional B and T cells are generated[26,27]. Similar to the observation on BALB/c mice, YB1 treatment significantly inhibited lung metastasis, while only slightly delayed primary tumor growth in the 4T1-NOD SCID orthotopic metastasis experiments (Fig. 4a). Similarly, significant suppression of metastasis was observed in the 4T1-NOD SCID experimental metastasis model (Fig. 4b). Also, colonization of cancer cells in the lungs of NOD SCID mice was impaired when mice were pretreated with YB1 (Fig. 4c). Altogether, these data demonstrated that the innate immune system is sufficient for *Salmonella* YB1-induced suppression of cancer cell metastasis.

**IFN-γ-dependent accumulation of neutrophils and NK cells in the lung of *Salmonella*-treated NOD SCID mice.** To identify which cells of the innate immunity contribute to the metastasis suppression, we examined IFN-γ-dependent changes of the innate immunity after *Salmonella* treatment in NOD SCID mice.

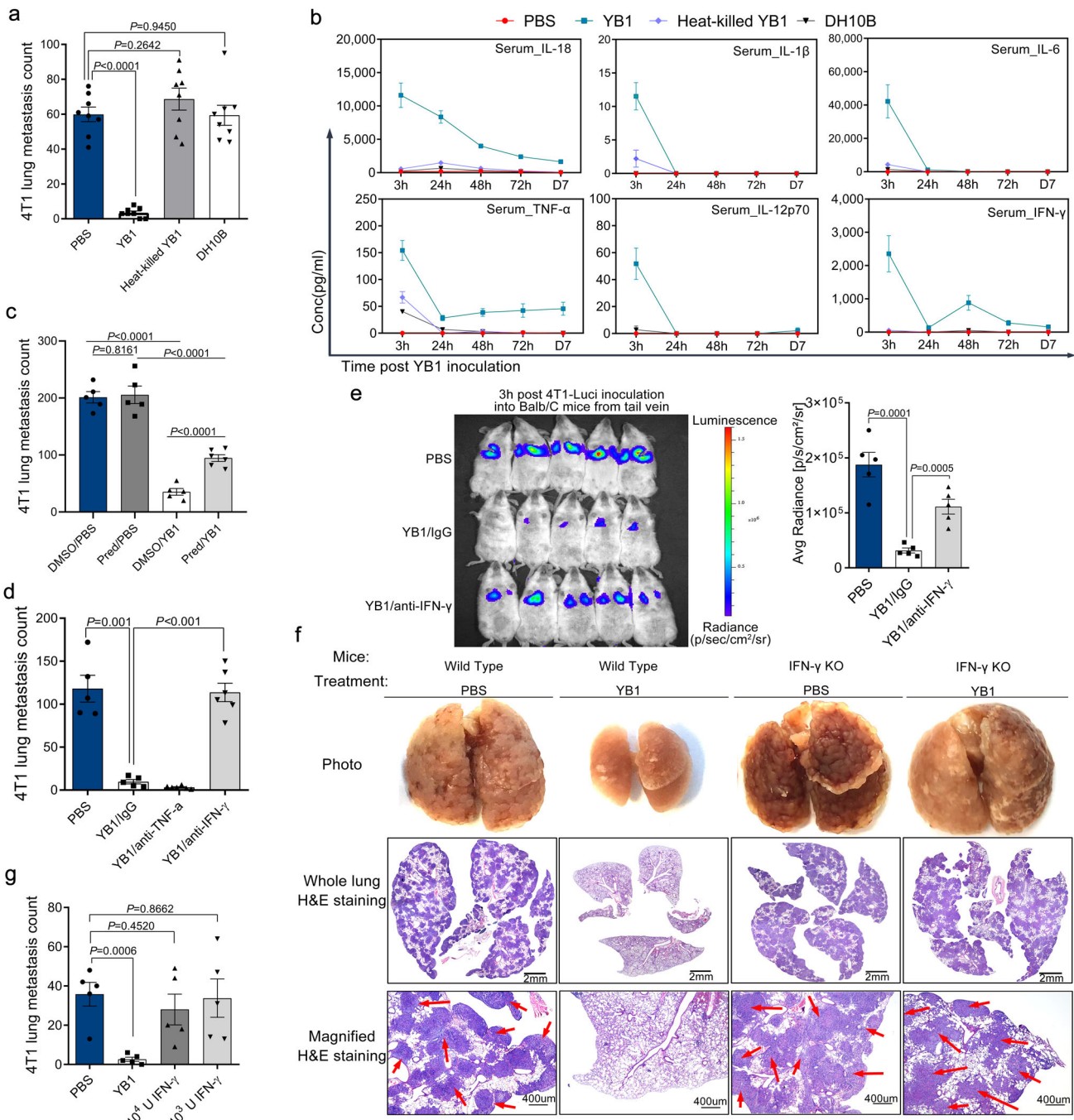

**Fig. 3 IFN-γ-dependent inflammation induced by *Salmonella* YB1 infection facilitates the suppression of cancer metastasis. a** Quantification (two-tailed unpaired *t*-tests, n = 8 per group) of 4T1 lung metastases in mice treated with PBS, YB1, heat-killed YB1, and *E. coli* DH10B, respectively, on day 0. 4T1 cancer cells were intravenously inoculated on day 13 and lung metastasis was quantified on day 27. Kinetics of cytokine levels in the serum of these BALB/c mice were monitored (*n* = 4 per group) (**b**). Conc, concentration. **c** Quantification (two-tailed unpaired *t*-tests) of 4T1 lung metastases after YB1-induced inflammation was inhibited by prednisolone in the 4T1-BALB/c experimental metastasis model (*n* = 6 Pred/YB1, *n* = 5 others). Pred, prednisolone. **d** Quantification (two-tailed unpaired *t*-tests) of 4T1 lung metastases following antibody-mediated depletion of TNF-α and IFN-γ in the 4T1-BALB/c experimental metastasis model (*n* = 5 for PBS and YB1/IgG; *n* = 6 for YB1/anti-TNF-α and YB1/anti-IFN-γ). **e** Tracking and quantification (two-tailed unpaired *t*-tests) of 4T1-Luci cells in vivo by luciferase live imaging 3 h after i.v. injection of 4T1-Luci into BALB/c mice pretreated with PBS, YB1 plus isotype IgG, or YB1 plus IFN-γ depletion antibody, respectively. BALB/c mice were pretreated 3 days in advance with PBS or YB1 (*n* = 5 per group). **f** Comparison of the anti-metastatic activity of YB1 in wild-type C57BL/6J mice and IFN-γ-knockout (IFN-γ KO) mice based on the experimental metastasis model established with MB49 cancer cells (*n* = 5 mice per group). Representative lung pictures, H&E-stained whole lung, and magnified H&E-stained lung tissue from each group were shown. Red arrows indicate partial tumor tissues. Plenty of MB49 tumor nodules were found in lungs from IFN-γ-knockout mice treated either with PBS or YB1 and wild-type mice treated with PBS. **g** Quantification of 4T1 lung metastases (two-tailed unpaired *t*-tests) after treatment with recombinant IFN-γ compared to YB1 or PBS treatment. This experiment was based on the 4T1-BALB/c experimental metastasis model (*n* = 5 per group). All data are presented as mean values ± SEM. All displayed are one representative experiment of two independent experiments. Source data are provided as a Source Data file.

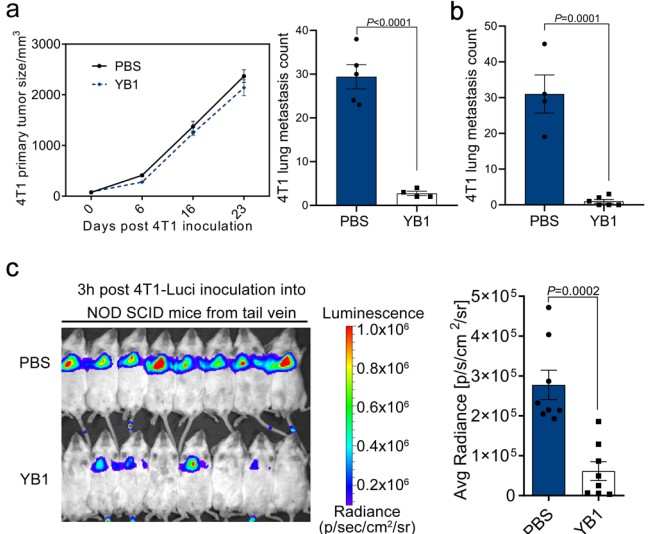

**Fig. 4 The innate immune responses are sufficient for *Salmonella* YB1 to inhibit cancer metastasis. a** Quantification of 4T1 primary tumor size and relevant lung metastases (two-tailed unpaired *t*-test) after YB1 treatment (*n* = 4) in the 4T1-NOD SCID orthotopic metastasis model. PBS as a control treatment (*n* = 5). **b** Quantification (two-tailed unpaired *t*-test) of 4T1 lung metastases in the 4T1-NOD SCID experimental metastasis model with YB1 treatment (*n* = 4 for PBS group; *n* = 6 for YB1 group). **c** Tracking and quantification (two-tailed unpaired *t*-test) of 4T1-Luci cells in vivo by luciferase live imaging 3 h after i.v. injection of 4T1-Luci into NOD SCID mice pretreated 3 days in advance with either PBS or YB1 (*n* = 8 mice per group). The surface intensity of bioluminescence was measured with the region of interest (ROI) tools from Living image 4.0 software. All data are presented as mean values ± SEM. All displayed are one representative experiment of two independent experiments. Source data are provided as a Source Data file.

We used CyTOF, a high-dimensional state-of-the-art flow cytometry platform, to profile innate immune cells and their secretion of cytokines. Immune cells were isolated from the lungs of NOD SCID mice treated with PBS, YB1 (YB1/IgG control), and YB1 plus IFN-γ depletion antibody (YB1/anti-IFN-γ) (Fig. 5a). The body weight changes were monitored (Supplementary Fig. 5a). Notably, the total number of immune cells was greatly increased after the YB1 treatment compared to the PBS group, whereas IFN-γ depletion reduced the number of immune cells in the lung (Supplementary Fig. 5b). This indicates that IFN-γ might promote the accumulation of immune cells in the lung. The staining panel used in the CyTOF analysis includes a broad range of phenotypic markers and cytokines that allow detailed classification of the subtypes of innate immune cells, as well as characterization of their active states (Supplementary Table 1). We performed t-Distributed Stochastic Neighbor Embedding (t-SNE) dimension reduction and PhenoGraph clustering analyses of the CyTOF data, to characterize the lung-infiltrating immune cells, which identified 18 immune cell clusters.

We analyzed the profiles of the 18 immune cell clusters across all samples (Supplementary Fig. 5c). Figure 5b shows a representative t-SNE profile of each group. Clustering analysis grouped the samples from the same treatment together, validating the t-SNE dimension reduction and PhenoGraph clustering analyses (Supplementary Fig. 5d). To compare the distribution and abundance of the 18 identified immune cell clusters among the three treatment groups, we summarized the percentage and total cell number of each cluster across samples (Fig. 5c, d). As we suspected the IFN-γ-dependent innate immune changes induced by *Salmonella* YB1 treatment could contribute to the metastasis

suppression, we focused on changes that were induced by YB1 but restored by IFN-γ depletion. We observed that clusters 13, 14, 15, and 18 were highly upregulated after YB1 treatment, but were restored to almost the levels in the control group after IFN-γ depletion. To characterize each immune cell cluster, we pooled the CyTOF data from all samples and analyzed the expression levels of surface markers and cytokines in each cell cluster (Fig. 5e). Clusters 13, 14, and 18 were identified to be three NK subtypes (CD3− NKp46+ CD49b+) and cluster 15 was a neutrophil subtype (CD45+ CD11b+ Ly6G^high Ly6C^low). Five NK cell subtypes (cluster 2, 13, 14, 16, and 18) were detected in the lungs of NOD SCID mice, in which NK cell activity was previously thought to be markedly reduced[26]. Among the 5 NK subtypes, clusters 13, 14, and 16 were thought to be more active because of higher expressions of CD38 and CD44, which indicated enhanced activation and migration of immune cells[28–30]. Besides, cluster 13 is the only NK cell type that produces IFN-γ. Given that IL-10-secreting neutrophils have immunosuppressive activity in melanoma patients[31], the neutrophil cluster 15, which was characterized to produce high levels of IL-10 and with low expression of cell activation marker CD38, might act as a negative regulator of the immune responses induced by *Salmonella* and barely contribute to its anti-metastatic efficacy. Altogether, we revealed IFN-γ-dependent accumulation of four immune cell clusters, including three NK subtypes (cluster 13, 14, and 18) and one neutrophil subtype, which might be involved in the anti-metastatic activity of YB1.

***Salmonella*-provoked NK cells are integral to cancer metastasis suppression.** To confirm the roles of neutrophils and NK cells in suppressing metastasis, we depleted neutrophils and NK cells using specific antibodies. Injection of anti-Ly6G antibody every other day depleted more than 90% of peripheral neutrophils[32] (Supplementary Fig. 6a, b), but the *Salmonella* treatment was still able to inhibit metastasis, indicating that neutrophils were not the key immune cells involved in YB1-induced metastasis suppression (Supplementary Fig. 6c). Injection of anti-Asialo-GM1 antibody was able to deplete NK cells in BALB/c mice, as previously described, with a minor modification[33,34]. The NK depletion efficiency was confirmed by the absence of peripheral CD3− NKp46+ cells (Fig. 6a, b). Gating strategies for live NK cell identification and later relevant assays were addressed in Supplementary Fig. 6d–f. The depletion of NK cells abolished the metastasis suppression by YB1 and even led to more metastatic nodules in the lungs compared to the PBS group (Fig. 6c). We previously showed that YB1 induced the elimination of lung accumulated cancer cells (Fig. 2g–i). Here we showed that NK cell depletion could totally abolish this YB1-induced effect (Fig. 6d, e). The off-target effect of anti-Asialo-GM1 antibody has been reported in 2011 and the major concern is that basophils are also affected[35]. Basophils represent about 0.4% of circulating white blood cells. We then examined their existence in immunocompetent (BALB/c) and immunodeficient (NOD SCID and NSG (NOD.Cg-Prkdc^scid Il2rg^tm1Wjl/SzJ)) mice. We first examined the percentage of basophils in the lung of BALB/c mice 6 days after YB1 treatment. In contrast to the dramatic change of NK cells after YB1 treatment, the percentage of basophils did not show a significant difference after YB1 treatment (Supplementary Fig. 7a, b). Also, basophils only account for around 0.2% of all immune cells in the lung (Supplementary Fig. 7a, b). We further examined the existence of basophils and NK cells in NOD SCID and NSG mice. Basophils were hardly detected in NOD SCID mice and NSG mice (<0.05%), whereas NK cells only exist in BALB/c mice and NOD SCID (Supplementary Fig. 7c–e). Together with our data showing that YB1 can potently inhibit

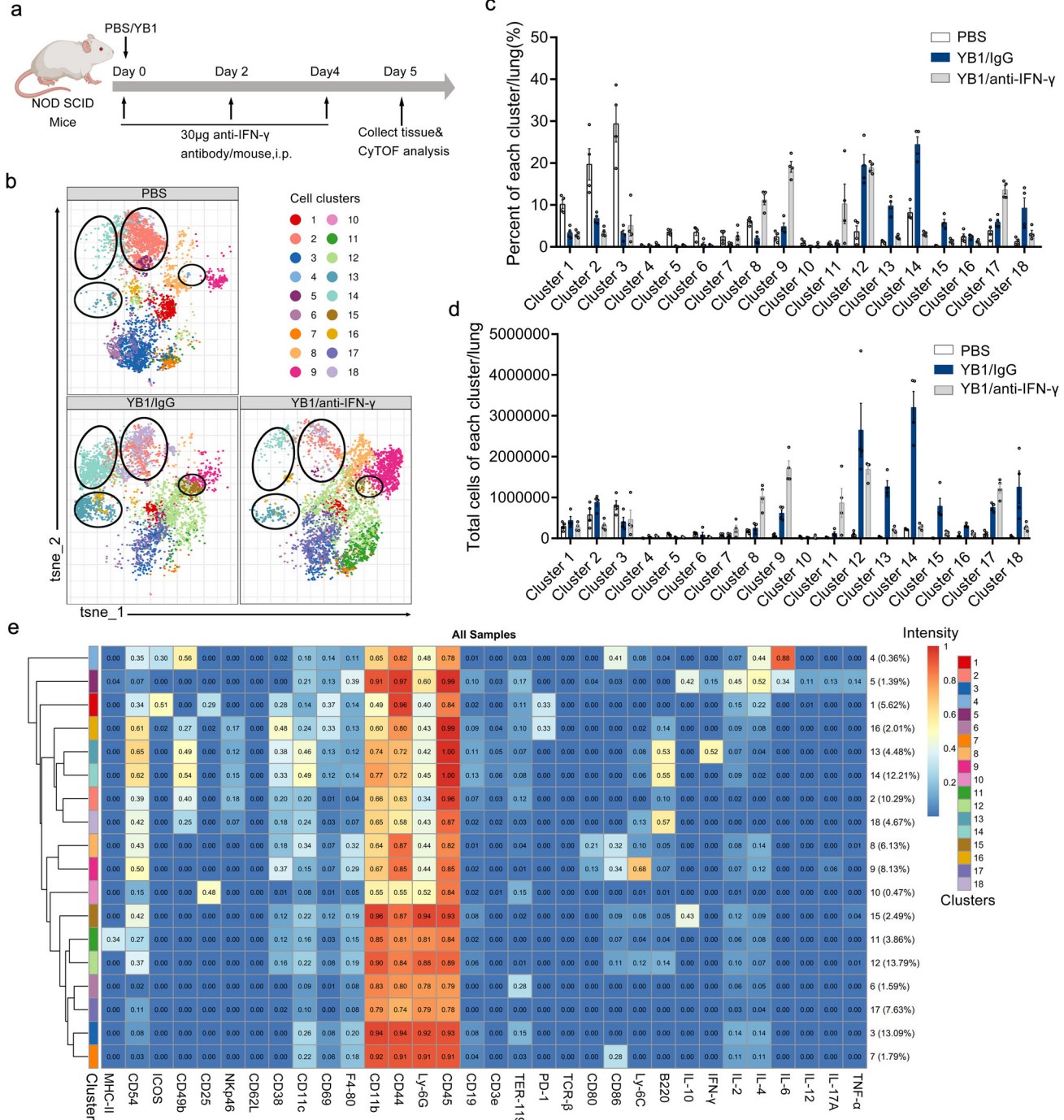

**Fig. 5 IFN-γ promotes the accumulation of neutrophils and NK cells in the lung environment of NOD SCID mice after *Salmonella* YB1 treatment. a** Mice treatment timeline for CyTOF analysis. NOD SCID mice were divided into three groups treated with PBS, YB1 (YB1/IgG), or YB1 plus IFN-γ depletion antibody (YB1/anti-IFN-γ), respectively. All mice were killed on day 5 and lung-infiltrating immune cells were isolated and prepared for CyTOF analysis. **b** Representative t-SNE profile from each group of mice. Immune cell clusters 13, 14, 15, and 18 are highlighted by circles. **c** Percentage of each immune cell cluster per lung across samples (*n* = 4 per group). **d** Total cell number of each immune cell cluster per lung across samples (*n* = 4 per group). Immune cell clusters 13, 14, 15, and 18 were upregulated in the YB1 group compared to the other two control groups. **c**, **d** All data are presented as mean values ± SEM. **e** Characterization of each immune cell cluster. Based on the expressions of different phenotypic markers and cytokines, events acquired by CyTOF analysis of lung-infiltrating immune cells from NOD SCID mice with the three different treatments were merged for the t-SNE dimension reduction and PhenoGraph clustering analyses, and were divided into 18 immune cell clusters. Displayed one representative experiment of two independent experiments for all panels. Source data are provided as a Source Data file.

metastasis in NOD SCID mice (Fig. 4), we could conclude that basophils are not likely to be the key immune cells mediating the anti-metastasis effect of *Salmonella* YB1. Alternative to antibody-mediated NK cell depletion, we also tested the anti-metastatic

ability of YB1 in NSG mice. NSG mice are generated by crossing NOD SCID mice and IL2rg−/− mice, which resulted in the complete lack of NK cells, T cells, and B cells, but with functional neutrophils[36]. EGFP-labeled 4T1 cells (4T1-EGFP; $5 \times 10^4$) were

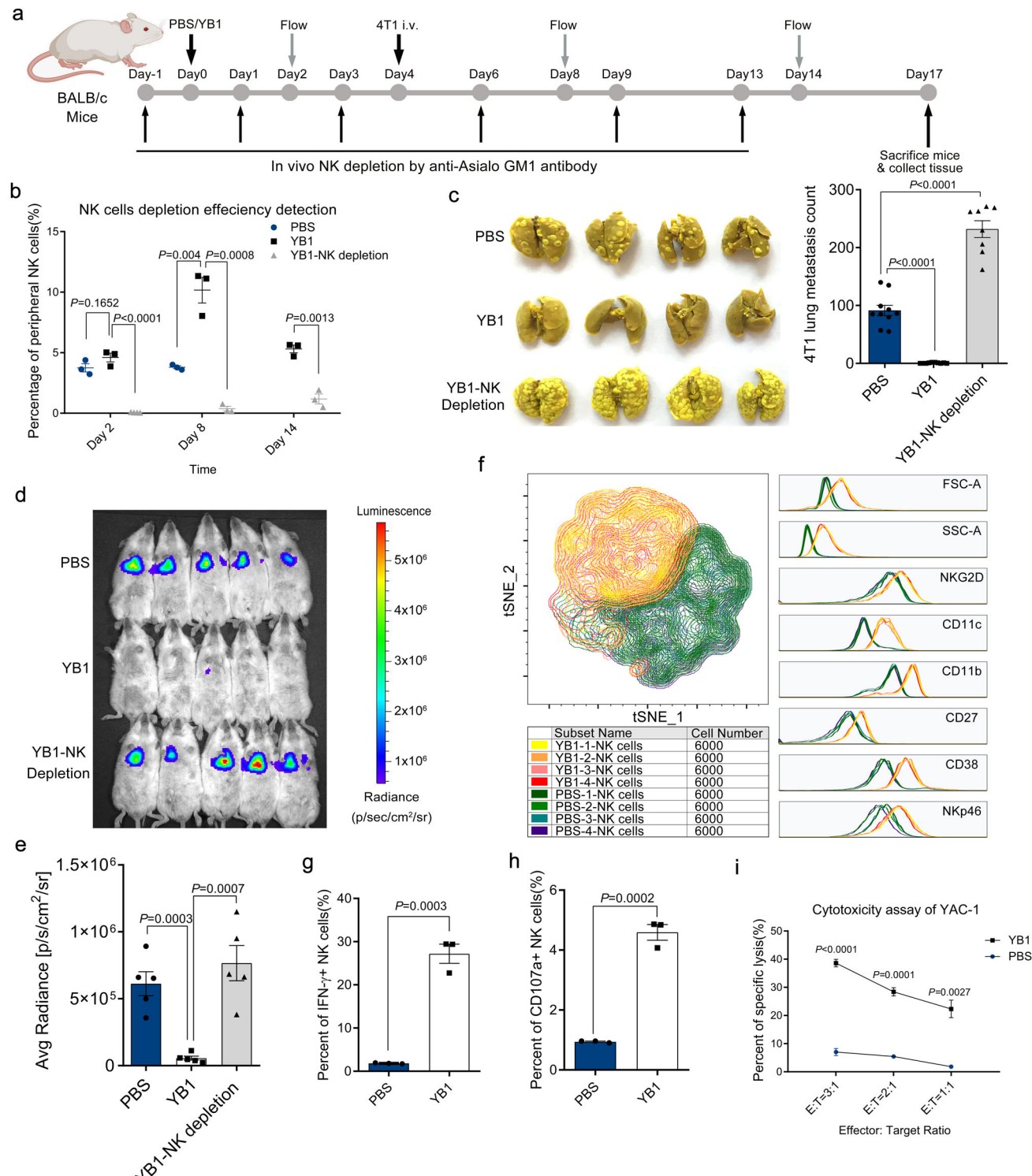

i.v. injected into NSG mice, to establish lung metastasis. Similar to the antibody depletion results, YB1 failed to significantly inhibit lung metastasis in the NSG mice model (Supplementary Fig. 7f). Altogether, the results indicate NK cells, but not neutrophils or basophils, are essential for *Salmonella* YB1-induced metastasis suppression.

In the present study, we defined CD3-NKp46+ cells as classical NK cells, but CD3-NKp46+ cells also include ILC1s, a rare non-cytotoxic innate lymphoid cell type that is capable of producing IFN-γ and shares many markers with classical NK cells across

tissues[37]. In addition to the cytotoxicity, expression of Eomes can also be used to distinguish NK cells from ILC1s[38]. However, we found more than 96% of Lin-NKp46+ cells (CD3-NKp46+ cells included) are Eomes+, indicating they are classical cytotoxic NK cells. Moreover, the Lin-NKp46+ Eomes− cells (defined as ILC1s) did not show a higher ability to produce IFN-γ (Supplementary Fig. 8a–c). We further examined the levels of granzyme B and perforin in lung-infiltrating CD3-NKp46+ immune cells from mice treated with *Salmonella* YB1 and found that CD3-NKp46+ cells from YB1-treated mice have significantly

**Fig. 6 *Salmonella* YB1-provoked NK cells are integral to the cancer metastasis suppression. a** BALB/c mice were treated with PBS, YB1, or YB1 plus NK cell depletion antibody, respectively. NK cell depletion efficiency was validated on days 2, 8, and 14 by flow cytometry. 4T1 cancer cells were i.v. implanted on day 4. **b** Percentage of peripheral blood NK cells at the indicated time points across three groups ($n = 3$ mice per group). On day 14, PBS group samples were unavailable. **c** Picture and quantification ($n = 10$ for PBS or YB1 group; $n = 8$ for YB1-NK depletion group, combined results of 2 independent experiments) of 4T1 lung metastases across three groups. **d, e** Tracking and quantification of 4T1-Luci cells in vivo by luciferase live imaging 3 h after i.v. injection of 4T1-Luci into BALB/c mice pretreated 3 days in advance with PBS, YB1, or YB1 plus NK depletion antibody, respectively ($n = 5$ per group). **f** Overlay of t-SNE maps of lung-infiltrating NK cells from mice either pretreated with PBS or YB1 5 days before lung organ harvest ($n = 4$ per group). After flow data acquisition, 6000 NK cells were randomly selected from each sample for the t-SNE analysis in the FlowJo software. NK cells from mice treated with PBS or YB1 had two distinct NK cell subtypes according to FSC (forward scatter), SSC (side scatter), and expressions of different markers (CD11b, CD11c, CD38, NKG2D, NKp46, and CD27). **g** Flow cytometric analysis of IFN-γ production in lung-infiltrating NK cells after co-culture with PMA and Ionomycin ex vivo for 5 h ($n = 3$ per group). **h** Flow cytometric analysis of CD107a expression on lung-infiltrating NK cells after co-culture with YAC-1 cells ex vivo for 5 h ($n = 3$ per group). **i** Cytotoxicity of lung-infiltrating NK cells against YAC-1 target cells at the indicated NK: YAC-1 (E : T, Effector : Target) ratios ($n = 3$ biological replicates). Lung-infiltrating immune cells were isolated from mice 5 days post treatment. All data are presented as mean values ± SEM. All *P*-values were derived using two-tailed unpaired *t*-tests. Displayed one representative experiment of two independent experiments, except for **c**. Gating strategies were addressed in Supplementary Fig. 6d–f. Source data are provided as a Source Data file.

higher levels of granzyme B and perforin (Supplementary Fig. 8d, e). These results together indicate that most CD3-NKp46+ cells after YB1 treatment, if not all, are conventional cytotoxic NK cells. Although we cannot fully rule out the possibility of ILC1s in *Salmonella*-provoked metastasis suppression, the current data have not provided strong support for it. Using flow cytometry, we identified an obvious phenotypic change in lung-infiltrating NK cells from mice with *Salmonella* YB1 treatment. Lung-infiltrating NK cells from mice treated with PBS or YB1 were isolated and analyzed for surface markers. We performed a t-SNE analysis and found that NK cells from PBS and YB1 groups formed two distinct populations (Fig. 6f). We found NK cells isolated from YB1-treated mice were larger in size (higher FSC (forward scatter), and SSC (side scatter)) and had higher expression levels of CD38 and NKG2D, suggesting an activated status. The YB1-activated NK cells also had high expression levels of CD11b and CD11c, consistent with NK cell clusters 13 and 14 identified by CyTOF in NOD SCID mice (Figs. 5c–e and 6f). Next, we compared the activity of NK cells isolated from mice treated with PBS or YB1. Production of IFN-γ and expression of CD107a are commonly regarded as functional markers for assessing cytokine production and degranulation of NK cells[39]. Indeed, NK cells from YB1-treated mice produced more IFN-γ and showed higher CD107a expression ex vivo (Fig. 6g, h). Cytotoxicity assay was performed on the isolated NK cells against YAC-1 cancer cells as previously described[34]. We assayed effector/target ratios of 3 : 1, 2 : 1, and 1 : 1, and found NK cells from YB1-treated mice had higher cytotoxicity against YAC-1 at all ratios (Fig. 6i), which is consistent with higher levels of granzyme B and perforin after YB1 treatment (Supplementary Fig. 8d, e). Altogether, the results showed YB1-provoked NK cells were larger, had greater cytotoxicity to tumor cells, and had higher expression levels of CD11b, CD11c, NKG2D, and CD38 compared to the NK cells from PBS-treated mice.

**NK cells and IFN-γ are indispensable for *Salmonella*-induced cancer metastasis suppression.** We showed that the anti-metastatic activity of YB1 requires both IFN-γ and NK cells. NK cells activated by *Salmonella* treatment produce a large amount of IFN-γ; however, whether IFN-γ could directly or indirectly act on NK cells remains unknown. To examine the effects of NK cells on the serum dynamics of IFN-γ, we treated mice with NK-cell depletion antibody to deplete NK cells, which was confirmed by the absence of peripheral CD3− NKp46+ cells (Fig. 7a, b). Serum IFN-γ was measured at the indicated time points after NK cell depletion (Fig. 7a, c). At 3 h, we did not detect any IFN-γ production in mice receiving NK cell depletion

antibody with YB1 treatment, whereas the concentration was very high in the YB1 treatment group without NK depletion, indicating NK cells are responsible for IFN-γ production in the early stage after YB1 treatment. However, on days 2–4, the IFN-γ concentration gradually increased even though NK cells were still absent, suggesting other cells produce IFN-γ at later stages after YB1 treatment (Fig. 7c). The plasma TNF-α showed quite similar dynamics as IFN-γ after NK depletion in vivo (Supplementary Fig. 8f). When 4T1-Luci cells were i.v. injected on day 4 to establish lung metastasis, accumulated cancer cells in the lung were not eliminated by YB1 treatment in the absence of NK cells, even with high levels of IFN-γ (Fig. 7c–e). Altogether, these results demonstrate that IFN-γ functions through NK cells to mediate metastasis suppression, and that NK cells are the major source of IFN-γ in the early stage of *Salmonella* YB1 infection, but IFN-γ can be later produced by other immune cells.

Lastly, we investigated whether IFN-γ depletion affects NK cell activation and cytotoxicity. We isolated lung-infiltrating NK cells 5 days after initial IFN-γ depletion in BALB/c mice treated with YB1. The IFN-γ depletion significantly reduced the amount of NK cells in the lung measured either by NK cell percentage or by total cell number (Fig. 7f, g). IFN-γ-producing NK cells when stimulated ex vivo were also remarkably reduced, indicating fewer NK cells were activated in the absence of IFN-γ (Fig. 7h). The expression level of NK cell degranulation marker CD107a was also decreased to the same level as in the PBS group after IFN-γ depletion in vivo (Fig. 7i). Consistently, the cytotoxicity of NK cells isolated from IFN-γ-depleted mice was largely decreased, as indicated by the YAC-1 lysis assay (Fig. 7j). These results imply that IFN-γ promotes the accumulation, activation, and, especially, cytotoxicity of NK cells upon *Salmonella* YB1 treatment.

Overall, *Salmonella* YB1 stimulation can trigger robust secretion of IFN-γ by NK cells during the early stage of infection. However, IFN-γ is also produced by other cells 2 days after *Salmonella* infection, which is consistent with previous work[40,41]. In turn, the systemic high level of IFN-γ promotes the accumulation and/or activation of NK cells in the lung, as suggested by our finding that both IFN-γ and NK cells are required for *Salmonella*-induced suppression of cancer metastasis.

## Discussion

The development of "perfect" bacteria for cancer therapy relies greatly on our understanding of host–bacteria interactions. Unfortunately, not much is known about the anti-tumor or anti-metastatic ability of bacteria, besides the generally accepted idea that bacteria elicit host immune responses[12,42]. Bacterial infections initiate complex interactions between bacteria and the host

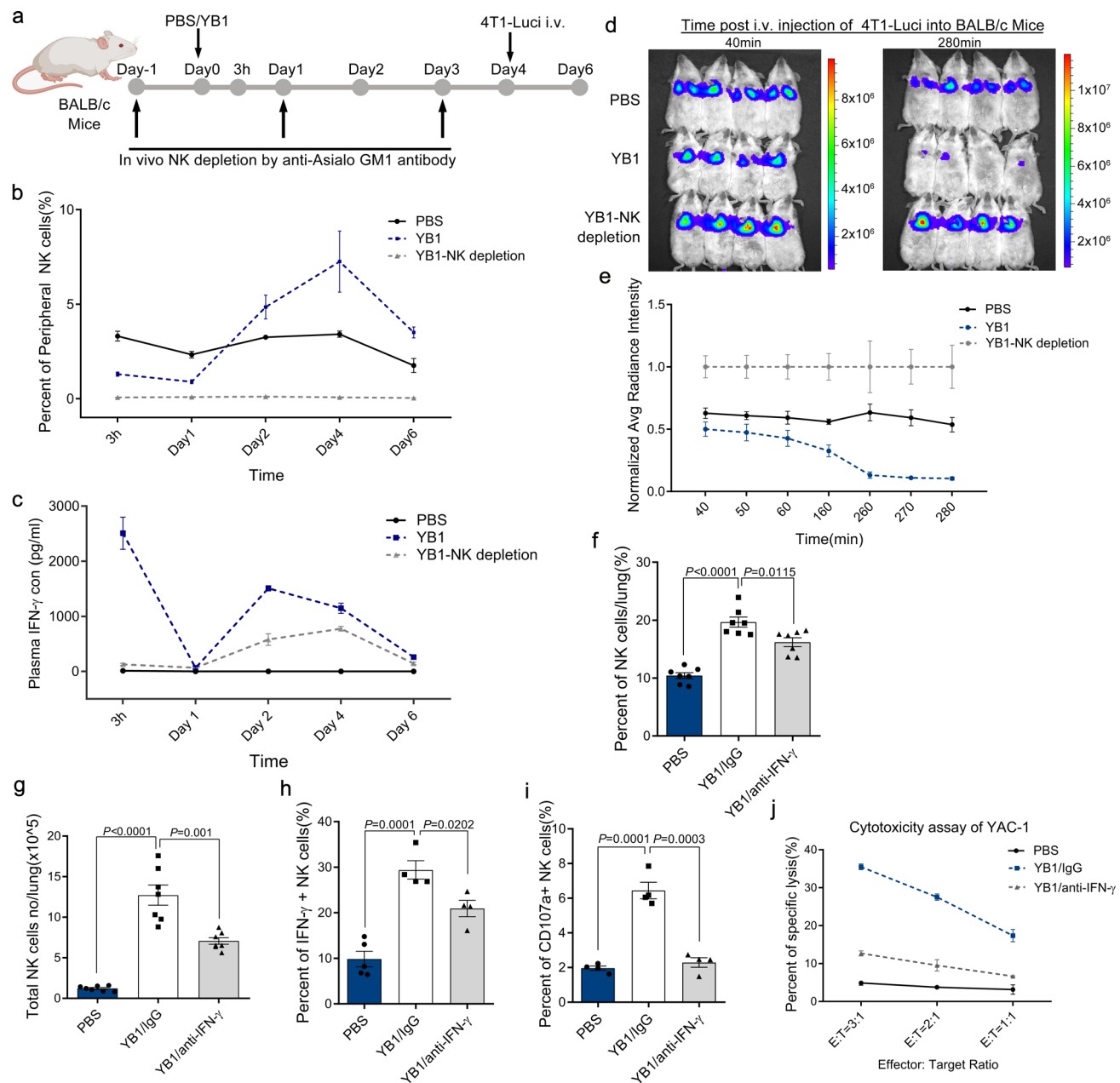

**Fig. 7 NK cells and IFN-γ are indispensable for *Salmonella* YB1 to suppress cancer metastasis. a** BALB/c mice were divided into three groups and treated with PBS, YB1, or YB1 plus NK depletion antibody, respectively (*n* = 4 per group). **b**, **c** NK cell depletion efficiency (*n* = 3 mice per group) and plasma IFN-γ concentration (*n* = 4 mice per group) were monitored at 3 h, day 1, 2, 4, and 6. **d**, **e** On the morning of day 4, 4T1-Luci cancer cells were i.v. injected into BALB/c mice and tracked by live imaging (*n* = 4 mice per group). Data were normalized to YB1-NK depletion group. **f**–**i** BALB/c mice were divided into three groups and treated with PBS, YB1 (YB1/IgG) or YB1 plus IFN-γ depletion antibody (YB1/anti-IFN-γ), respectively. All mice were killed on day 5 after YB1 or PBS treatment and lung-infiltrating immune cells were applied to flow cytometric analysis. **f** The percentage of NK cells to all immune cells in lung across samples (*n* = 7 per group). **g** The absolute total number of NK cells per lung across samples (*n* = 7 per group). Total immune cell numbers were measured by trypan blue exclusion and then multiplied by the percentage of NK cells determined by FACS analysis to give the absolute number of NK cells for each lung. **h** Flow cytometric analysis of IFN-γ production in lung-infiltrating NK cells across samples after co-culture with PMA and Ionomycin ex vivo for 5 h (*n* = 5 for PBS group, *n* = 4 for YB1/IgG and YB1/anti-IFN-γ groups). **i** Flow cytometric analysis of CD107a expression on lung-infiltrating NK cells across samples after co-culture with YAC-1 cells ex vivo for 5 h (*n* = 4 mice per group). **j** Cytotoxicity of lung-infiltrating NK cells against YAC-1 target cells at the indicated NK : Yac-1 (E : T, Effector : Target) ratios (*n* = 3 biological replicates). All data are presented as mean values ± SEM. All *p*-values were derived using two-tailed unpaired *t*-tests. Displayed is one representative experiment of two independent experiments, except for **f**, **g** (combined results of two independent experiments). Source data are provided as a Source Data file.

immune system, making it difficult to identify the key factors. Widely used as a DNA vaccine delivery system, *Salmonella* SL7207 showed promising suppressive effect on the growth of primary tumors, especially when carrying all kinds of therapeutic payloads[43,44]. However, non-negligible toxicity of *Salmonella*

SL7207 on the host dampened its further use for anti-tumor therapies[45]. YB1, a significant refinement on SL7207, retains the same anti-tumor potency but greatly reduces the reactogenicity of *Salmonella* in mice (Supplementary Fig. 1g–i). Based on the discovery that *Salmonella* YB1 suppressed metastasis in a liver

cancer model[7], we systemically studied the anti-metastatic ability of *Salmonella* (YB1 as a representative) to unravel the underlying mechanisms. Here we report that *Salmonella* has potent suppressive effects on cancer metastasis via the host innate immunity regardless of cancer type or host genetic background. Moreover, we have identified cytokine IFN-γ and NK cells as the key mediators in bacterial inhibition of cancer cell metastasis. We found that IFN-γ is the key cytokine for cancer metastasis suppression by engineered *Salmonella*, which is mainly mediated through activated NK cells.

Previous research demonstrated an important role of NK cells in the control of metastasis[46]. Regulated by integrated signals from activating and inhibitory ligands, as well as from cytokine (such as IL-12, IL-18, and IL-15), properly activated NK cells possess anti-metastatic activity independent of MHC (major histocompatibility complex)-mediated antigen presentation[46]. However, cancer cells have developed various strategies to escape recognition and attack by NK cells during metastasis, including the following: (i) modification of NK cell-activating ligands and NK cell inhibitory ligands[47–49]; (ii) recruitment of immunosuppressive cells, such as classical CD11b+Ly6G+ neutrophils, Treg cells, and platelets[50–52]; and (iii) secretion of immunosuppressive factors, such as transforming growth factor-β and IL-10[53–55]. Inspired by these inhibitory mechanisms, current investigations of NK cell-based immunotherapies have focused on the expansion and activation of NK cells by cytokines or by modification of NK cell receptors/ligands axes[56–58]. Here we demonstrated a different approach using engineered *Salmonella* to efficiently drive NK cells into a subtype with high anti-metastatic activity characterized by higher expression of phenotypic and activated markers including CD11b, CD11c, CD38, and NKG2D. In addition to the well-addressed IFN-γ production by NK cells, we also demonstrated a new phenomenon that IFN-γ in turn is essential for the accumulation, activation, and cytotoxicity of NK cells in vivo upon *Salmonella* treatment.

As a pleiotropic cytokine, IFN-γ is conventionally known to elicit potent anti-tumor immunity by modulating adaptive immune responses including, but not limited to, the upregulation of MHC class I (major histocompatibility complex class I), the polarization of Th1 immune responses, activation of CTL, and regulation of Treg cells[59,60]. In addition to the regulation of immune responses, IFN-γ was reported to directly function on cancer cells and endothelial cells, inducing senescence of cancer cells and regression of tumor vasculature[61–63]. Our study revealed a new role of IFN-γ in mediating bacteria-induced metastasis suppression by modulating the innate immune responses, especially the functions of NK cells. Interestingly, a simple application of recombinant IFN-γ failed to inhibit lung metastasis. A possible explanation could be that it is difficult to recapitulate the spatiotemporal kinetics of IFN-γ in vivo by i.v. injection of IFN-γ alone or to establish a positive-feedback loop between IFN-γ and activated NK cells. On day 5 after *Salmonella* treatment, we found that IFN-γ was decreased in the tumor but highly elevated in the lungs. An i.v. injection of IFN-γ might not be able to achieve a high-enough IFN-γ concentration in lung tissue (Supplementary Fig. 3f). The major source of IFN-γ during oral *Salmonella* infection was from neutrophils and macrophages, whereas NK cells secreted very little[23]. However, we found that during the early stage of i.v. *Salmonella* infection, NK cells were the dominant source of IFN-γ (Fig. 7c), indicating the activation of immune cells by *Salmonella* is dependent on the infection route.

Production of IFN-γ by activated NK cells has been well-studied and is used as a marker to evaluate NK cell status[39]. However, whether IFN-γ directly or indirectly affects the anti-metastatic function of NK cells remains unclear. A study of the orthopoxvirus infection suggests that macrophages, IFN-γ, and

CXCR3 are required for the recruitment of NK cells[64]. IFN-γ was also reported to counterbalance the immunosuppressive environment of the lung mediated by NK cells in IFN-γ[−/−] mice during *Mycoplasma* infection[65]. We further showed that upon depletion of IFN-γ in vivo, the total number of NK cells and IFN-γ-producing NK cells in the lung were significantly reduced during the *Salmonella* treatment (Fig. 7g, h), and cytotoxicity of NK cells was also greatly reduced (Fig. 7j). These data imply a role of IFN-γ in enhancing the cytotoxicity of NK on cancer cells.

Although our results demonstrate that the enhanced anti-metastatic effect of NK cells in vivo is dependent on IFN-γ, it is possible that other regulators also contribute to the cytotoxicity of NK cells upon *Salmonella* treatment. For example, cytokines such as IL-2, IL-12, IL-15, IL-18, IL-21, and type I IFNs are positively involved in the maturation, activation, and survival of NK cells[66,67]. Among these cytokines, we observed IL-18 was systemically increased after the *Salmonella* treatment. In addition, dysfunction of platelets has been documented to facilitate the clearance of circulating cancer cells by NK cells[52]. Furthermore, how IFN-γ promotes the accumulation, activation, and cytotoxicity of NK cells in vivo remains to be elucidated upon *Salmonella* treatment. Other innate immune cells such as neutrophils and macrophages may be positively or negatively involved in this process.

The major players in the anti-metastatic immune response include cytotoxic T cells and NK cells[68,69]. Tumors can escape the recognition of cytotoxic T cells by downregulation or mutation of MHC class I molecules, which limits the therapeutic application of T-cell-based immunotherapies, such as immune checkpoint blockade[70,71]. In contrast, the recognition of cancer cells by NK cells does not require neoantigens or tumor-associated antigens, or prior sensitization. In addition, loss of MHC class I expression on cancer cells increases their susceptibility to the killing by NK cells[72]. Therefore, engineering bacteria to enhance the cytotoxicity of NK cells would provide a promising strategy against cancer metastasis. Further research on the detailed mechanisms of NK cell activation and the role of IFN-γ in engineered *Salmonella* treatment will maximize the ability of NK cells and IFN-γ in treating metastatic disease.

## Methods

**Bacterial strains and cell lines**. *S. typhimurium* strain YB1 was engineered in our laboratory[6] and *E. coli* strain DH10B was a common laboratory strain. The YB1 was cultured in LB (luria-bertani) medium supplemented with DAP (100 µg/mL, Sigma D1377-5G), chloramphenicol (25 µg/mL), and streptomycin (50 µg/mL); DH10B was cultured in LB medium supplemented with streptomycin (50 µg/mL). Mouse 4T1 breast cancer cells, CT26 colon cancer cells, and B16F10 melanoma cell line were purchased from ATCC; Murine YAC-1 cells were kindly provided by Stem Cell Bank, Chinese Academy of Sciences; L1210-GFP cells were generously provided by Dr. Gaoliang Ouyang, Xiamen University; MB49 cancer cells were generously provided by Dr. LIU Chenli, Shenzhen Institute of Synthetic Biology. 4T1-EGFP cell line was constructed by chromosomal integration of an EF1α-EGFP cassette with a transposon system provided by Dr. Wei Huang; 4T1-Luci and MB49-Luci cell lines were generated with the transposon system harnessing firefly luciferase. The 4T1, CT26, and YAC-1 cells were maintained in RPMI 1640 medium supplemented with 10% fetal bovine serum, streptomycin, and penicillin in a tissue incubator at 37 °C and 5% CO₂. The B16F10 cells, L1210-GFP cells, MB49 cells were maintained in Dulbecco's modified Eagle's medium supplemented with 10% fetal bovine serum, streptomycin, and penicillin in a tissue incubator at 37 °C and 5% CO₂.

**Animals**. The 6- to 8-week-old female BALB/c, NOD.Cg-Prkd[cscid]Il2rg[tm1Wjl]/SzJ (NSG), and NOD.CB17-Prkdc[scid]/J (NOD SCID) mice were purchased from the Laboratory Animal Unit of The University of Hong Kong. The 6- to 8-week-old male and female C57BL/6J mice were purchased from the Laboratory Animal Unit of Shenzhen Institutes of Advanced Technology Chinese Academy of Sciences. The IFN-γ knockout male and female mice (JAX stock #002287) were purchased from The Jackson Laboratory and maintained in the Laboratory Animal Unit of Shenzhen Institutes of Advanced Technology Chinese Academy of Sciences. Mice were maintained in a 12 h light/12 h dark cycle at ~23 °C and 40% relative humidity. All mice experiments conducted in Hong Kong were approved by the

Committee on the Use of Live Animals in Teaching and Research of The University of Hong Kong. All animal experiments conducted in Shenzhen complied with protocols approved by the Shenzhen Institutes of Advanced Technology Chinese Academy of Sciences Committee on Animal Care.

**Orthotopic and experimental metastasis models.** For 4T1-BALB/c orthotopic metastasis model, 4T1 cells grown to 50–80% confluency were collected and washed three times with sterile PBS. Then 4T1 cells were prestained with 0.4% trypan blue and the number of viable cells was determined using a hemacytometer. 4T1 cells were diluted to $2 \times 10^6$/mL in PBS and the resuspended cells (100 μL) were injected into the mouse fat pad using a syringe with a 27 G needle. For NOD SCID orthotopic metastasis model, $3 \times 10^4$ 4T1-EGFP cells were injected into the fat pad to establish the primary tumor. Tumor size was measured using calipers and was calculated as $4/3 \times \pi \times (\text{height} \times \text{width}^2)/8$. To establish the experimental metastasis model, $1 \times 10^5$ 4T1 cells or $5 \times 10^4$ 4T1-EGFP cells resuspended in 100 μL PBS were i.v. injected into each BALB/c or NOD SCID/NSG mice, respectively. For CT26 experimental metastasis model, $1 \times 10^5$ CT26 cells were i.v. injected into BALB/c mice. For B16F10 experimental metastasis model, $2 \times 10^5$ B16F10 melanoma cells were i.v. injected into C57BL/6J mice. Metastasis in the above models was quantified by counting lung metastatic nodules under a stereomicroscope. For MB49 experimental metastasis models, $8 \times 10^5$ MB49 cells resuspended in 100 μL PBS were i.v. injected into C57BL/6J or IFN-γ-knockout mice.

**Bacterial treatment.** All bacteria used in the treatments were prepared from overnight culture. Cell numbers were determined by measuring OD at 600 nm (1 OD = $1 \times 10^9$ colony-forming units (CFU)). Bacteria adjusted to the desired concentration were washed three times in sterile PBS. For BALB/c mice experiments, $2 \times 10^7$ CFU of YB1, heat-killed YB1, and DH10B were i.v. injected through the tail vein. For the NOD SCID, NSG, C57BL/6J, and IFN-γ-knockout mice experiments, $8 \times 10^6$ CFU of YB1 bacteria were i.v. injected through the tail vein.

**CFU test.** Tissues including the lung, liver, spleen, and tumor were dissected at the indicated time points after *Salmonella* treatment. The samples were cut into small pieces and homogenized in sterile PBS using a microtube homogenizer (Sigma). Blood was collected in heparin-coated tubes and subject to fivefold series dilutions in 0.1% EDTA solution. The dilutions (10 μL) were applied to an agar plate containing DAP, streptomycin, and chloramphenicol to detect YB1. For each sample, two to three dilutions were chosen and each dilution was repeated four times.

**Immunohistochemistry.** Tissues were fixed in 4% paraformaldehyde in PBS for 24 h at 4 °C with shaking. After embedding in a paraffin block, 5 μm-thick sections were prepared and stained with hematoxylin and eosin (H&E). Antigen retrieval was performed in sodium citrate buffer (pH = 6.0) in a water bath at 100 °C for 20 min. Tissue sections were then blocked with blocking solution (2% bovine serum albumin (BSA) and 5% normal serum in PBS) for 30 min. Rabbit anti-E-cadherin and rabbit anti-vimentin antibodies (Cell Signaling) diluted in 1% BSA in PBS were applied to different sections. Horseradish peroxidase-conjugated secondary antibody in combination with DAB peroxidase substrate (Dako) were used to visualize the staining. Sections were counterstained with hematoxylin and analyzed using a BX51 light microscope.

**Luciferase live imaging.** Eight-week-old female BALB/c or NOD SCID mice were i.v. injected with YB1 on day 0. On day 3, $6 \times 10^5$ 4T1-luci cells or $3 \times 10^5$ 4T1-luci cells were i.v. injected through the tail vein of BALB/c or NOD SCID mice separately. For C57BL/6J mice, YB1 bacteria were i.v. injected into mice on day 0, followed by $2 \times 10^6$ MB49-luci cells i.v. injected through the tail vein on day 3. For in vivo imaging, mice were anesthetized with ketamine and medetomidine, followed by 100 μL D-luciferin (30 mg/mL, Gold Biotechnology) intraperitoneally (i. p.) injected 5 min before live imaging. Images were taken with the PE IVIS Spectrum in vivo imaging system and analyzed with Living image 4.0 software. The surface intensity of bioluminescence was measured with region of interest tools from Living image 4.0 software.

**Cytokine quantification.** Blood was collected from the tail vein and serum was stored at −80 °C. Lung and tumor were freshly dissected, minced, weighed before transferring into the tissue lysis buffer (10 mM Tris-HCl, 150 mM NaCl, 1% NP-40, 10% Glycerol, 5 mM EDTA, and Protease inhibitor cocktail). Tissue samples were homogenized using a microtube homogenizer (Sigma) and incubated for 1 h at 4 °C with gentle shaking. The lysate was sonicated with a 5 min regimen of 9 s on and 9 s off using a SONICS vibra cell sonicator (amplitude, 40%). Sonicated samples were centrifuged at 15,000 × g for 20 min at 4 °C. The supernatant was collected and aliquots were stored at −80 °C. Cytokines in the prepared samples were quantified using either a mouse Th1/Th2 10plex FlowCytomix Multiplex kit (eBioscience, BMS820FF) or a customized ProcartaPlex Multiplex immunoassay panel (eBioscience) according to the manufacturer's instructions. Elisa Kit (ThermoFisher Scientific, KMC4022) was used for serum IFN-γ quantification. Elisa kit (R&D, MTA00B) was used for plasma TNF-α quantification.

**Immunosuppression.** On days −2, −1, 0, 1, 2, and 3, prednisolone (10 mg kg$^{-1}$ day$^{-1}$) in 55% dimethylsulfoxide (DMSO) or 55% DMSO alone was i.p. injected into mice. On day 0, YB1 or PBS was administrated to the designated groups. On day 5, 4T1 cells were i.v. injected into mice. All the mice were killed on day 19 to count metastatic nodules and for H&E staining.

**Flow cytometry.** For cell surface markers detection, immune cells were stained with antibodies for 30 min on ice and washed twice with 1% BSA/PBS before flow cytometric analysis. For intracellular detection of IFN-γ, immune cells were induced ex vivo with PMA/ionomycin supplemented with brefeldin A for 5 h. After culture, cell mixtures were collected and stained for cell surface markers. Following fixation and permeabilization of cells (BD, catalog number 554714), cells were stained with anti-IFN-γ antibody on ice for 30 min and washed twice before flow cytometric analysis. BD LSR Fortessa Analyzer, ACEA Novocyte Flow Cytometer, FlowJo_v10.6.2, and NovoExpress_v1.4.1 were used for the flow cytometric analysis.

**Antibody for flow cytometry.** All antibodies used for flow cytometry analysis were listed in Supplementary Table 2. All gating strategies were addressed in the Supplementary Fig. 6d–f.

**Antibody-mediated depletion of cytokines.** Antibodies were used to neutralize TNF-α and IFN-γ cytokines. On day 0, BALB/c mice were i.v. injected with YB1 or PBS. On days 0, 2, and 4, the YB1-treated mice were i.p. injected with antibodies against TNF-α (1.25 mg/kg, diluted in 100 μL DPBS; #16-7423-81; clone TN3-19.12; eBioscience) or antibodies against IFN-γ (1.25 mg/kg, diluted in 100 μL DPBS; #16-7411-85; clone H22; eBioscience). Anti-mouse IgG (#16-4888-38; eBio299Arm; eBioscience) was used as the control. On day 6, $1 \times 10^5$ 4T1 cells were i.v. injected through the tail vein to establish lung metastasis. After 2 weeks, all mice were killed to examine lung metastasis by counting lung metastatic nodules under a stereomicroscope. For flow cytometric analysis, mice were killed on day 5 after YB1 treatment and immune cells were isolated for flow cytometric analysis.

**IFN-γ treatment.** On day 0, BALB/c mice were i.v. treated with PBS, YB1, and $10^3$ or $10^4$ U of recombinant IFN-γ ($6 \times 10^6$ U/mg, diluted in 200 μL DPBS; BioLegend), respectively. On day 6, 4T1 cells were i.v. injected through the tail vein to establish lung metastasis. After 2 weeks, all mice were killed to compare lung metastasis by counting metastatic nodules.

**Isolation of organ-infiltrating immune cell.** Spleen, LNs, and lung were collected from mice. To obtain single-cell suspensions from the spleen or LN, spleen or LN tissue was passed through a 200-gauge mesh and then washed with PBS. To obtain single-cell suspensions from tumor and lung, Collagenase Type IV (Sigma-Aldrich, catalog number C5138) and DNase I Type IV (Sigma-Aldrich, catalog number D5025) were used to dissolve minced tissue chunks at 37 °C. After lysis of red blood cells, single-cell suspensions were resuspended in 40% Percoll and carefully overlaid on 70% Percoll. Immune cells were enriched from the interface of 40%/70% Percoll after density gradient centrifugation at 1260 × g for 30 min at room temperature.

**Mass cytometry (CyTOF) analysis.** Unconjugated antibodies were labeled using Fludigm's labeling kit according to the manufacturer's instructions. Briefly, cells were stained with 0.75 μM cisplatin (Fluidigm) for 2 min at room temperature and blocked with Fc-receptor blocking buffer (Biolegend). After blocking, cells were incubated with the antibody cocktail on ice for 40 min for staining cell surface makers. Next, cells were fixed in 1× Maxpar Fix I Buffer (Fluidigm) and then washed three times with Maxpar Perm-S Buffer (Fluidigm), followed by intracellular staining of cytokines for 30 min on ice. After washing, cells were incubated in 1 mL intercalator buffer (0.125 nM MaxPar Intercalator-Ir in 1 mL Fix and Perm buffer). Cells were then washed with deionized water and resuspended with EQ Four Element Calibration Beads (Fluidigm) immediately before data acquisition. Events were acquired using CyTOF2 (Version 6.7.1014, Fluidigm) at an event rate of <500 events/second. Mass cytometry data were normalized to EQ 4-element bead signal (Lot P15K0802, Passport EQ 4_P13H2302) in 100 s interval windows using normalization software version 2 (Version 6.7.1014, Fluidigm). Mass tag barcodes were resolved with a doublet filtering scheme using Debarcoder (Fluidigm). Live immune cells were manually gated in FlowJo by event length, live/dead discrimination, and the desired expression of CD45. Data were then exported for downstream analysis and transformed with a coefficient of 5 with method cytofAsinh. For most downstream analyses, the individual sample data were subsampled to 5000 events (or all cells if the total cell number was <5000). All samples were at least 2000 events in the final. t-SNE dimension reduction and PhenoGraph clustering analyses were performed using the tool cytofkit run in R. All markers were used during the t-SNE and PhenoGraph analyses. For the generation of heatmap displays, marker expression was normalized by dividing by the range of all markers (expression range from 1 to 99 percentile).

**NK cell depletion in vivo**. To validate the role of NK cells on the anti-metastatic effect induced by *Salmonella* YB1, NK cells were depleted in vivo by i.p. injection with 30 μL anti-Asialo-GM1 (#146002; clone Poly21460; BioLegend) antibody on day −1, 1, and 3. NK depletion was maintained by i.p. injection of 20 μL anti-Asialo-GM1 antibody on days 7, 10, and 13. During the experiments, YB1 was injected on day 0 and cancer cells were i.v. injected on day 4. Matched isotype rabbit polyclonal IgG (#BE0095; polyclone; BioXcell) served as the control. NK depletion was confirmed by the absence of CD3− NKp46+ cells in the peripheral blood.

**Neutrophil in vivo depletion**. To deplete neutrophils in vivo, 200 μg anti-Ly6G antibody (#BP0075-1; clone 1A8; BioXCell) was i.p. injected 1 day before YB1 injection and maintained by i.p. injection of 50 μg anti-Ly6G antibody every other day until the end of the experiment. Matched isotype rat IgG2A (#BP0089; clone 2A3; BioXCell) served as the control. Neutrophil depletion efficiency was confirmed by the detection of peripheral neutrophils.

**Degranulation assay**. Lung-infiltrating immune cells were co-cultured with 0.4× the number of YAC-1 cells labeled with cell trace violet for 5 h at 37 °C in 5% $CO_2$. GolgiStop (BD) and anti-CD107a (BD) were supplemented during the culture. After culture, cell mixtures were collected and stained for cell surface markers and subjected to flow cytometric analysis.

**Purification and culture of murine NK cells**. Murine NK cells were collected from the lung. Lymphocytes were isolated from the lung by density gradient centrifugation with 40% and 70% Percoll. Murine NK cells were purified using NK Cell Isolation Kit (Miltenyi Biotec, catalog number 130-115-818) according to the manufacturer's instructions. Purified NK cells were resuspended in Iscove's modified Dulbecco's medium supplemented with 10% fetal calf serum, L-glutamine (1 mM; Gibco), streptomycin (100 μg/mL; Sigma), penicillin (100 IU/mL; Sigma), and 50 μM β-mercaptoethanol.

**Flow cytometry-based killing assay**. YAC-1 target cells labeled with cell trace violet (ThermoFisher, catalog number C34571) were co-cultured with effector NK cells in 96-well plate at various effector: target (E/T) ratios for 4 h at 37 °C. The maximum killing was achieved by incubating YAC-1 target cells with an equal volume of absolute ethanol. After culturing, cell mixtures were collected and washed for flow cytometric analysis. Propidium iodide-positive YAC-1 cells were defined as dead YAC-1 cells. The percentage of specific lysis was calculated for each well as follows:

$$\% \text{ of specific lysis} = 100 \times (\text{Dead Ratio}^{\text{experimental}} - \text{Dead Ratio}^{\text{spontaneous}})/(\text{Dead Ratio}^{\text{maximumkilling}} - \text{Dead Ratio}^{\text{spontaneous}})$$

**Statistical analysis**. Data were analyzed using GraphPad Prism version 8.0.1. The analyses are indicated in corresponding legends. *P*-values are indicated in corresponding figures.

**Reporting summary**. Further information on research design is available in the Nature Research Reporting Summary linked to this article.

## Data availability

All data supporting the findings of this study are available within the article and its Supplementary Information files, and from the corresponding author upon reasonable request. Source data are provided with this paper.

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

## Acknowledgements

We thank Dr. Wei Huang for providing the transposon system for the construction of cell lines. We thank Dr. Jing Guo and Ms. Yau Ka Long for the technical support for flow cytometry. We thank Dr. Pentao Liu for the critical suggestions on the project. Icon for mouse from Figs. 1a, d, 5a, 6a, 7a, and Supplementary Figs. 1d and 3d was created with BioRender.com under a paid subscription hold by Badea SR. This work was supported by grants from the Shenzhen Peacock Team Project (KQTD2015033117210153), the National Natural Science Foundation of China (number 91957120), the Shenzhen Science and Technology Innovation Committee Basic Science Research Grant (JCYJ20170413154523577), the National Key R&D Program of China (2018YFA0902701), the National Natural Science Foundation of China (numbers 32001040, 31730029, and 31770952), and the Program for Guangdong Introducing Innovative and Enterpreneurial Teams (2019BT02Y198). J.D.H. is supported by the L&T Charitable Foundation. The funding bodies did not contribute to the design of the study, or collection, analysis, and interpretation of the data.

## Author contributions

The work presented here was carried out through the collaboration of all authors. J.D.H. directed and coordinated this study. Q.B.L., L.R., and J.D.H. designed the experiments and analyzed the data. Q.B.L., L.R., X.J., B.Y., R.H.L., and C.X. conducted the experiments. M.C.L., N.Z., and H.R.G. helped design parts of the experiments. Z.B.Z., G.F.F., and K.J.L. were involved in mice experiments. G.F., X.J., and X.L.C. directed the CyTOF experiment and summarized the CyTOF results. X.L. and J.C.H. helped analyze the CyTOF results. S.H.L. provided critical comments on the project. and Q.B.L., L.R., and J.D.H. wrote the manuscript. All authors read, commented on, and approved this manuscript.

## Competing interests

The authors declare no competing interest.
