## [Peer Review File · Nature Communications]

Reviewers' comments:

Reviewer #1 (Expertise: Salmonella, cancer, Remarks to the Author):

This is an interesting paper concerning the inhibition of metastases by *Salmonella typhimurium*. One key to successful cancer therapy is to inhibit the formation of secondary tumours.

In an experimental murine model, a novel *Salmonella* where the *asd* gene is under control of a hypoxic promoter induces a response which appears to block the formation of secondary cancers. The timing of the introduction of the bacterium is critical. It is easy to see how an *asd* modification will have significant effects on the biology of the bacterium, especially as it moves between oxygenated and anaerobic environments. It is taken on faith that the YB1 strain has the phenotype claimed, there is no data showing this, and some early in the paper data showing the effect of the oxygen environment on growth of the bacterium would have supported claims that the effects observed were due to special properties of YB1.

The *aroA* mutant SL7207 is likely to be mouse lethal when given at a high dose, but much less so at a lower dose. At a lower dose does SL7207 have the same metastasis inhibition activity properties as YB1?

The core phenomenon looks very robust and appears to work across mouse strains with different metastatic tumour types. This is impressive.

The exact mechanism behind the inhibition was explored in some well conducted and sequential experiments. The use of antibody to inhibit IFN γ and antibodies (anti-Asialo GM1) that deplete NK cells suggest that IFN γ is central to the phenomenon. Such studies do not reveal whether the cytokine acts directly to promote killing by the NK cells, or whether it is part of a cascade e.g. ending with proteins such as granzymes or perforins. While this may be very difficult to resolve to specific mechanism, looking at effector molecules downstream of IFN γ might be informative. Use of an IFN γ Receptor KO might also be informative. Also, it could be important to show, somehow that the other cell types removed by anti-Asialo-GM1 antibody did not play a role in the phenomenon (e.g. basophils).

There is some in vitro killing by the *Salmonella* activated NKs of YAC1 targets but the ratios of NKs to targets seem high (1:1?). Are such high ratios likely to be physiological? It would have been nice too

to see whether the Salmonella-activated NKs had any lytic activity against a autologous tumour line. This may be unfeasible, but it would be helpful to understand such killing occurs.

I think the penultimate section of the results (lines 360-362) is only one explanation. The high levels of IFN γ might recruit and activate NKs, OR the NKs might be activated directly by Salmonella (e.g. through their inflammasome by Salmonella constituents, Kupz 2014 PMID: 24827856) to release a high level of IFN γ that can be detected in the plasma.

The paper is very well written – the prose is excellent and the experiments are logical.

More specific comments

Figure 1 needs to include the death curves (extended F1 C) present in the extended version – these are key

While there is evidence that NKs are removed by depletion, is it clear that the functional cytokine level is reduced. This should be shown for TNF depletion too, at least as a Supp Figure.

Assays such as ICS/CyTOF suggest that specific cell types can make the cytokine, but do not reveal whether it is secreted. There are assays that can do this (e.g. diabody-based FACS).

Reviewer #2 (Expertise: NK cells, cancer metastasis, Remarks to the Author):

The authors showed that the injection of engineered Salmonella YB1 can inhibit the development of lung metastasis in orthotopic and experimental metastasis model. YB1 interferes with the colonization of the lung by cancer cells. IFN- γ is required during this anti-metastatic process by promoting the accumulation and the cytotoxicity of NK cells in the lung.

This newly described role of IFN- γ as regulator of NK cell functions is interesting but several major concerns need to be addressed.

Major comments:

- The authors indicate the number of mice used in each experiment, but they do not indicate how many independent experiments they performed. The way the data are shown, and the number of used mice suggest that each experiment was performed only once, which is not sufficient and biologically relevant to properly conclude.
- The authors indicate that data were analyzed using GraphPad Prism but they never indicate what statistical tests are used. Without this information, the reviewer cannot appreciate the relevance of the indicated p-value.
- The orthotopic model using 4T1 cells is a correct model of metastasis development. It is interesting to show in parallel the experimental model of metastasis using the same cells. It is also informative to use other cells and mouse strains to give a global conclusion about the positive effect of YB1 on lung metastasis. But this reviewer deplores the fact that the other models are experimental metastasis models and not orthotopic metastasis models which would have had a better relevance.
- The authors defined NK cells as CD3- NKp46+ but this definition also includes ILC1s. Even if ILC1s are a minor population in the lung at steady state, as they are also depleted after anti-asialoGM1 treatment, it could be interesting to discriminate both populations. The reviewer is particularly intriguing by the cluster 16 and 18 of the figure 5 which are NKp46+ but CD49b low compared to other NKp46+ clusters. As these clusters are increased in term of cell number after YB1 treatment, a better characterization of what the authors called "NK cells" could be done in order to determine if ILC1s, in addition of NK cells, have a role in the observed anti-metastatic effect of YB1.
- The authors showed that IFN-g is involved in YB1-induced anti-metastatic effect. The reviewer wonders what is the source of this IFN-g. In figure 7, the authors showed that IFN-g is produced by other immune cells than NK cells after YB1 treatment in the last time point. The figure 4 showing that adaptive immune system is not required for YB1-induced metastasis suppression suggests that T cells are not the source of IFN-g. As the authors mentioned in the discussion, neutrophils and macrophages are the main source of IFN-g during oral infection by Salmonella. It would be interesting to know if this is also the case in this model.

Minor comments:

- The authors should be consistent across the text and decide if there is, or not, a space between the last word of the sentence and the number of the bibliographic reference
- Figure 1F: what is the length of the scale bar?
- Line 203: the authors probably forgot to delete "as well as"
- Extended figure 4C: what is the time of collection of the tumors?

- Line 342-344: the authors should gain clarity if they precise in the text that the 4T1-Luci cells were injected in day 4 post YB1 treatment, which corresponds to a high IFN-g serum concentration even in NK cell depleted mice.

- Line 675: Inappropriate capital letter on "Day"

We thank all the reviewers for their careful reading of our manuscript and for their insightful comments and suggestions. A point-by-point response to the reviewers' comments is given below.

Reviewer #1 (Expertise: Salmonella, cancer, Remarks to the Author):

This is an interesting paper concerning the inhibition of metastases by *Salmonella typhimurium*. One key to successful cancer therapy is to inhibit the formation of secondary tumours.

1) In an experimental murine model, a novel *Salmonella* where the *asd* gene is under control of a hypoxic promoter induces a response which appears to block the formation of secondary cancers. The timing of the introduction of the bacterium is critical. It is easy to see how an *asd* modification will have significant effects on the biology of the bacterium, especially as it moves between oxygenated and anaerobic environments. It is taken on faith that the YB1 strain has the phenotype claimed, there is no data showing this, and some early in the paper data showing the effect of the oxygen environment on growth of the bacterium would have supported claims that the effects observed were due to special properties of YB1.

Response:

The parental strain (of YB1) SL7207 also have anti-metastasis effect as shown below in Data 1 and 2 (also in Supplementary Fig. 1g in manuscript). Therefore, the anti-metastasis effect shall not be attributed to special properties of YB1. We also have other gene-knockout strains based on SL7207, such as *ssaT* and *InvA* knockout attenuated strains, and they all showed anti-metastasis effects as shown below in Data 1.

Data 1: Quantification of 4T1 lung metastases after treatments with different bacterial strains (n = 6 mice per group). DH10B is a common laboratory *E. coli* strain.

2) The *aroA* mutant SL7207 is likely to be mouse lethal when given at a high dose, but much less so at a lower dose. At a lower dose does SL7207 have the same metastasis inhibition activity properties as YB1?

Response:

When given at a lower dosage and followed by ampicillin treatment to control the *Salmonella* infection, SL7207 showed similar metastasis inhibition activity properties as YB1. The relevant data was added to Supplementary Fig. 1g and was shown below:

Data 2: Quantification (unpaired t-tests) of lung metastases in the 4T1-BALB/c experimental metastasis model after *Salmonella* SL7207 treatment. Ampicillin was given to both groups of mice from day 3 to kill *Salmonella* (n=5 in PBS group, n=6 in SL7207 group) and continued for 4 days.

3) The core phenomenon looks very robust and appears to work across mouse strains with different metastatic tumour types. This is impressive.

Response:

All tumor metastasis models we tried so far showed consistent results and YB1 can potentially inhibit metastasis of all tumor types tested.

4) The exact mechanism behind the inhibition was explored in some well conducted and sequential experiments. The use of antibody to inhibit IFN γ and antibodies (anti-Asialo GM1) that deplete NK cells suggest that IFN γ is central to the phenomenon. Such studies do not reveal whether the cytokine acts directly to promote killing by the NK cells, or whether it is part of a cascade e.g. ending with proteins such as granzymes or perforins. While this may be very difficult to resolve to specific mechanism, looking at effector molecules downstream of IFN γ might be informative. Use of an IFN γ Receptor KO might also be informative. Also, it could be important to show, somehow that the other cell types removed by anti-Asialo-GM1 antibody did not play a role in the phenomenon (e.g. basophils).

Response:

We also noticed the limitations of antibody-mediated depletion of cytokines and immune cells. We have conducted further experiments to facilitate our understanding of the interactions between IFN- γ and NK cells. We found that IFN- γ was mainly produced by NK cells during early *Salmonella* infection, and in turn, IFN- γ promoted the accumulation, activation and

cytotoxicity of NK cells. As for the effect of IFN- γ on the cytotoxicity of NK cells, our data suggest that IFN- γ shall be a part of a cascade ending with increased proteins such as granzymes or perforins.

First of all, we quantified the granzyme B and perforin of NK cells from PBS- or YB1-treated mice. The results showed that much higher ratio of NK cells from YB1 treated mice expressed granzyme B and perforin, as shown below in Data 3.

Data 3: YB1 treatment increased the granzyme B and perforin levels in NK cells. BALB/c mice were divided into two groups and treated with PBS or 2×10^7 YB1. All mice were sacrificed on day 6 after treatment and lung infiltrating immune cells were isolated and co-cultured with YAC-1 cells *ex vivo* for 5 h. After co-culturing, flow cytometric analysis of granzyme B (a) and perforin (b) in NK cells was performed (n=3 biological replicates). P-values are generated using unpaired t-tests. Displayed is one representative experiment of 2 independent experiments.

When IFN- γ was further depleted in YB1-treated mice, the ratio of NK cells that express granzyme B and perforin are also down regulated (Data 4), indicating that IFN- γ might induced a cascade response in NK cells which lead to elevated levels of granzyme B and perforin.

Data 4: Depletion of IFN- γ down regulated the levels of granzyme B and perforin in NK cells. BALB/c mice were divided into three groups and treated with PBS, 8×10^6 YB1

(YB1/IgG), or 8×10^6 YB1 plus IFN- γ depletion antibody (YB1/anti-IFN- γ), respectively. All mice were sacrificed on day 5 after YB1 or PBS treatment and lung infiltrating immune cells were isolated for flow cytometric analysis of NK cells. Flow cytometric analysis of granzyme B (a) and perforin (b) on lung infiltrating NK cells across samples after co-culture with YAC-1 cells *ex vivo* for 5 h was performed (n=3 biological replicates). P-values are generated using unpaired t-tests. Displayed is one representative experiment of 2 independent experiments.

Anti-Asialo-GM1 antibody is now widely applied to deplete NK cell *in vivo* and has been shown to have robust depletion effect^{1,2}. Nevertheless, off-target effect of anti-Asialo-GM1 antibody has been reported in 2011 and the major concerns are about basophils and some subpopulations of NKT cell, CD8+ T cell and $\gamma\delta$ T cell³. Our data show that *Salmonella* YB1 can still inhibit cancer metastasis in immunodeficient nude and NOD SCID mice (Data 5). Nude mice have defect in thymus formation, thus lacking functional NKT cell, CD8+ T cell and $\gamma\delta$ T cells^{4,5}. NOD SCID mice also have no functional CD8+ T cell⁶, which has been confirmed by our CyTOF data in Fig. 5e. Therefore, NKT cell, CD8+ T cell and $\gamma\delta$ T cell shall not play a major role in metastasis suppression induced by YB1.

Data 5: quantification (unpaired student's t-tests) of 4T1 lung metastases after YB1 treatment in nude mice and NOD SCID mice based on experimental metastasis model (n=5 mice per group). Displayed is one representative experiment of 2 independent experiments.

As for basophils, which represent about 0.4% of circulating white blood cells, we set up experiments to examine their existence in immunocompetent (BALB/c) and immunodeficient mice (NOD SCID and NSG). We firstly examined the percentage of basophils in the lung of BALB/c mice 6 days after YB1 treatment. In contrast to the dramatic change of NK cells after YB1 treatment, the percentage of basophils doesn't show significant difference after YB1 treatment. Also, basophils are quite rare in the lung and only account for around 0.2% of all immune cells, as shown below in Data 6 a, b (Supplementary Fig. 7a, b in manuscript). We further examined the existence of basophils and NK cells in NOD SCID and NSG mice. Basophils were hardly detected in NOD SCID mice and NSG mice (<0.05%), while NK cells only exist in BALB/c mice and NOD SCID, as shown in Data 6 c-e (Supplementary Fig. 7c-e in manuscript). Together with our previous data showing that YB1 can still potently inhibit metastasis in NOD SCID mice (Fig. 4 in manuscript and Data 5), which contain very few basophils (Data 6 c-e), we could conclude that basophils are unlikely to be a key immune cells

mediating anti-metastasis effect of *Salmonella* YB1.

Data 6: Basophils are not key immune cells mediating anti-metastasis effect of *Salmonella* YB1. **a** Representative flow cytometry plots of CD3-NKp46⁺ NK cells and CD3-CD49b⁺ FcεRI⁺ basophils in lung from mice treated as indicated (n=4 mice per group). Lung tissues were collected on day 6 after treatment. **b** Quantification of CD3-NKp46⁺ NK cells and CD3-CD49b⁺ FcεRI⁺ basophils mentioned in (a). **c** Representative flow cytometry plots of CD3-NKp46⁺ NK cells and CD3-CD49b⁺ FcεRI⁺ basophils in BALB/c mice, NOD SCID mice and NSG mice (n=3 mice per group). Blood samples are collected from three strains of mouse for flow cytometry analysis. **d-e** Quantification of NK cells (d) and basophils (e) mentioned in c. **f** Photos of 4T1-EGFP lung metastasis 12 days after *i.v.* injection of 5×10^4 4T1-EGFP cells into NSG mice. Overgrowth of 4T1-EGFP lung metastasis was found in all NSG mice treated with either PBS or YB1. All *p*-values were derived using two-tailed unpaired student's *t*-tests. All

data shown as the mean +/- s.e.m. * $P < 0.05$, ** $P < 0.01$, *** $P < 0.001$, **** $P < 0.0001$.
Displayed is one representative experiment of 2 independent experiments. Source data are provided as a Source Data file.

5) There is some *in vitro* killing by the Salmonella activated NKs of YAC1 targets but the ratios of NKs to targets seem high (1:1?). Are such high ratios likely to be physiological? It would have been nice too to see whether the Salmonella-activated NKs had any lytic activity against an autologous tumour line. This may be unfeasible, but it would be helpful to understand such killing occurs.

Response:

The ratios of NK cells to targets are close to the physiological condition. According to Physiological Data Summary of BALB/cJ (000651) mice (Table 1 as below), the concentration of white blood cells in the blood circulation of an 8 weeks' old mouse is around 5.4×10^3 cells/uL. According to our flow analysis, NK cells at least account for 3% of all white blood cells in blood circulation. Considering a 22g mouse has around 1.5ml of blood, the total number of NK cells in blood circulation should be more than 2.43×10^5 . Also, based on our flow cytometry analysis, there are at least 3×10^5 NK cells in the lung, and the number could be over 1×10^6 in mice treated with YB1, shown below as Data 7 (Fig. 7f, g in manuscript). We only injected 1×10^5 4T1 cancer cells into each mouse to establish lung metastasis. Therefore, at least in early stage when injected cancer cells did not replicate to a larger number, ratios of NK cells to targets (from 3:1 to 1:1) during *in vitro* killing assay should be physiological.

As for the target cells used for *in vitro* flow cytometry-based assessment of cytotoxicity of NK cells, NK sensitive tumor cell lines, such as human K562 cells, Daudi cell, Raji cell and mouse YAC-1 cells, are commonly used, rather than autologous tumor cell lines⁷⁻⁹. The reason is that autologous tumor cell lines are usually not sensitive to the killing of NK cells when performing *in vitro* assays, even though they can be efficiently killed by NK cells *in vivo*¹⁰. We speculate that somehow the *in vitro* culture system still cannot completely unleash the cytotoxicity of NK cells (the *in vitro* stimulation might not be sufficient). We have also tried to use 4T1 cancer cells to test the cytotoxicity of NK cells *in vitro* using flow cytometry based cell viability assay, and it turned out that most cancer cells stayed alive (PI negative) even though they were incubated with NK cells for several hours. The protocol we used to test the cytotoxicity of NK cells, especially the E:T ratios and the choice of target cells, is quite standard and also adapted by most researchers^{1,2,9}.

[redacted]

Table 1. Physiological Data Summary of BALB/cJ (000651) mice from The Jackson Laboratory.
Weblink: http://jackson.jax.org/rs/444-BUH-304/images/physiological_data_000651.pdf

Data 7: the percentage and absolute number of isolated NK cells from lung. **a** The percentage of NK cells to all immune cells in the lung across samples (n=4 mice per group). **b** The absolute total number of NK cells per lung across samples (n=4 mice per group). Total immune cell numbers were measured by trypan blue exclusion and then multiplied by the percentage of NK cells determined by FACS analysis to give the absolute number of NK cells for each lung.

6) I think the penultimate section of the results (lines 360-362) is only one explanation. The high levels of IFN γ might recruit and activate NKs, OR the NKs might be activated directly by *Salmonella* (e.g. through their inflammasome by *Salmonella* constituents, Kupz 2014 PMID: 24827856) to release a high level of IFN γ that can be detected in the plasma.

Response:

We agree with the reviewer. The possibility that NK cells might also be activated directly by *Salmonella* cannot be excluded here. But still, we have demonstrated *in vivo* that high level of IFN- γ promotes the accumulation and activation of NK cells. As shown in Fig. 7c, *Salmonella* can trigger the release of IFN- γ by NK cells 3 hours post infection, as the NK depletion group doesn't have such a burst of IFN- γ in plasma. However, at later stage after infection, other cells will also secrete IFN- γ together with NK cells, as indicated by the partial increase of IFN- γ in plasma of the NK depletion group at later stage after infection, which is consistent with previous report. In turn, the whole systemic high level of IFN- γ will promote the accumulation, activation, and especially the cytotoxicity of NK cells, leading to the elimination of tumor cells effectively few days after infection. We have revised this part in the manuscript and it is also shown as below.

Overall, *Salmonella* YB1 stimulation can trigger robust secretion of IFN- γ by NK cells during the early stage of infection. But, IFN- γ is also produced by other cells 2 days after *Salmonella* infection, which is consistent with previous work^{11,12}. In turn, the systemic high level of IFN- γ promotes the accumulation and/or activation of NK cells in the lung, as suggested by our finding that both IFN- γ and NK cells are required for *Salmonella*-induced suppression of cancer metastasis.

7) The paper is very well written – the prose is excellent and the experiments are logical.

Response: We thank the reviewer for her/his comment.

More specific comments:

8) Figure 1 needs to include the death curves (extended F1 C) present in the extended version – these are key

Response:

Thanks for the kind suggestion. We have added the mentioned survival curves to Fig. 1c in manuscript.

9) While there is evidence that NKs are removed by depletion, is it clear that the functional cytokine level is reduced. This should be shown for TNF depletion too, at least as a Supp Figure.

Response:

Thanks for the kind suggestion. We added this data to Supplementary Fig. 7i in the manuscript and also shown below as Data 8.

Data 8. Plasma TNF- α concentration (n=4 mice per group) were monitored at day 2 and 7. The NK depletion *in vivo* decreased the plasma concentration of TNF- α in early stage (day 2) of *Salmonella* YB1 infection, but not later stages (day 7). The plasma TNF- α dynamics is similar with the plasma IFN- γ dynamics after NK depletion (Fig. 7c in manuscript). However, only depletion of IFN- γ abolishes the anti-metastasis effect of *Salmonella* YB1 (Fig. 3d in manuscript).

10) Assays such as ICS/CyTOF suggest that specific cell types can make the cytokine, but do not reveal whether it is secreted. There are assays that can do this (e.g. diabody-based FACS).

Response:

This is a good suggestion. We will try to apply these new assays to our future studies.

Reviewer #2 (Expertise: NK cells, cancer metastasis, Remarks to the Author):

The authors showed that the injection of engineered *Salmonella* YB1 can inhibit the development of lung metastasis in orthotopic and experimental metastasis model. YB1 interferes with the colonization of the lung by cancer cells. IFN- γ is required during this anti-metastatic process by promoting the accumulation and the cytotoxicity of NK cells in the lung.

This newly described role of IFN- γ as regulator of NK cell functions is interesting but several major concerns need to be addressed.

Major comments:

1) The authors indicate the number of mice used in each experiment, but they do not indicate how many independent experiments they performed. The way the data are shown, and the number of used mice suggest that each experiment was performed only once, which is not sufficient and biologically relevant to properly conclude.

Response:

Thank you for the kind reminder. We have added relevant information to each figure legend. For most of mice experiments, at least two independent experiments were conducted. For some of them, such as the metastasis inhibition, we have repeated for more than 5 times and the results are consistent.

2) The authors indicate that data were analyzed using GraphPad Prism but they never indicate what statistical tests are used. Without this information, the reviewer cannot appreciate the relevance of the indicated p-value.

Response:

Thank you again for your kind reminder. We have added information about statistical tests used for each figure in legend in the revised manuscript.

3) The orthotopic model using 4T1 cells is a correct model of metastasis development. It is interesting to show in parallel the experimental model of metastasis using the same cells. It is also informative to use other cells and mouse strains to give a global conclusion about the positive effect of YB1 on lung metastasis. But this reviewer deplores the fact that the other models are experimental metastasis models and not orthotopic metastasis models which would have had a better relevance.

Response:

We used several experimental metastasis models to investigate the colonization process of cancer cells interfered by *Salmonella* YB1. In the orthotopic model using 4T1 cells, we showed that the inhibition of metastasis by YB1 might happen in the early stages of metastasis, probably during the colonization process. As for the primary tumors, the inhibition effects are not significant compared to metastasis inhibition, therefore, experimental metastasis models are used for most of the time. Nevertheless, we do have evaluated the anti-metastasis activity of YB1 in a liver orthotopic model, as shown below in Data 9¹³. In this model, luciferase-labelled hepatocellular carcinoma cell line MHCC-97L was used to establish primary tumor in the liver of nude mice. Similar to the result from 4T1 model, lung metastasis was greatly reduced after YB1 treatment.

Data 9: *Salmonella* YB1 inhibits lung metastasis in a HCC orthotopic liver tumor model. Two weeks after orthotopic liver tumor implantation, YB1 and saline were administered

through the tail vein of the mice respectively. (a) Lung metastasis were inhibited at 3 weeks after YB1 treatment shown by representative photos. Lung metastasis was monitored by live imaging with Xenogen IVIS. (b) Representative histological features of lung metastatic tumors 3 weeks after treatment (treatment, n=14; control, n=9). Hematoxylin and eosin staining; magnification, $\times 100$.

4) The authors defined NK cells as CD3- NKp46+ but this definition also includes ILC1s. Even if ILC1s are a minor population in the lung at steady state, as they are also depleted after anti-asialoGM1 treatment, it could be interesting to discriminate both populations. The reviewer is particularly intriguing by the cluster 16 and 18 of the Fig. 5 which are NKp46+ but CD49b low compared to other NKp46+ clusters. As these clusters are increased in term of cell number after YB1 treatment, a better characterization of what the authors called “NK cells” could be done in order to determine if ILC1s, in addition of NK cells, have a role in the observed anti-metastatic effect of YB1.

Response:

We understand that the CD3-NKp46+ cluster also includes non-NK ILC1s. The distinction between NK cells and non-NK ILC1s is a difficult one to make due to shared use of many markers across tissues^{14,15}. Furthermore, a criterion valid for discrimination of NK cells from non-NK ILC1s in one tissue does not appear to hold for other tissues¹⁶. Nevertheless, the cytotoxic property and expression of perforin and granzyme of NK cells can be used to distinct from non-NK ILC1s¹⁷.

[redacted]

We examined the levels of granzyme B and perforin in CD3-NKp46+ lung infiltrating immune cells from mice treated with *Salmonella* YB1. We found that CD3-NKp46+ cells from YB1 treated mice have significantly higher level of granzyme B and perforin, as shown below in Data 10 (Supplementary Fig. 7g-h in manuscript). This result indicates that most CD3-NKp46+ cells after YB1 treatment, if not all, are conventional cytotoxic NK cells. We cannot exclude the

“helper” effects of non-NK ILC1s here, as they might contribute to the anti-metastasis activity through the secretion of IFN- γ .

Data 10: CD3-NKp46⁺ immune cells from YB1-treated mice showed higher level of granzyme B and perforin. BALB/c mice were divided into two groups and treated with PBS or 2×10^7 YB1. All mice were sacrificed on day 6 after treatment and lung infiltrating immune cells were isolated for flow cytometric analysis of NK cells. Flow cytometric analysis of granzyme B (a) and perforin (b) on lung infiltrating NK cells across samples after co-culture with YAC-1 cells *ex vivo* for 5 h was performed (n=3 biological replicates).

5) The authors showed that IFN-g is involved in YB1-induced anti-metastatic effect. The reviewer wonders what is the source of this IFN-g. In figure 7, the authors showed that IFN-g is produced by other immune cells than NK cells after YB1 treatment in the last time point. The figure 4 showing that adaptive immune system is not required for YB1-induced metastasis suppression suggests that T cells are not the source of IFN-g. As the authors mentioned in the discussion, neutrophils and macrophages are the main source of IFN-g during oral infection by Salmonella. It would be interesting to know if this is also the case in this model.

Response:

Our study indicates that IFN- γ and NK cells are dependent on each other to achieve the anti-metastasis effect. In NOD SCID mice, we detected major IFN- γ production in NK cells (cluster 13) and minor IFN- γ production in macrophages (cluster 5) by CyTOF analysis (Fig. 5e) 5 days after YB1 treatment. In BALB/c mice, we found NK cells are the major source of IFN- γ at early stage of infection (first two days after infection), but not at later stage (Fig. 7c). Therefore, during the early infection stage NK cells produce most of IFN- γ , and later other immune cells such as macrophages can also produce IFN- γ . Regardless of the source of IFN- γ , NK cells are required to exert anti-metastasis activity of *Salmonella* YB1.

Minor comments:

6) The authors should be consistent across the text and decide if there is, or not, a space between the last word of the sentence and the number of the bibliographic reference.

Response:

We have gone through the manuscript and made the mentioned format consistent across the text.

7) Figure 1F: what is the length of the scale bar?

Response:

We have added scale bar in relevant legend and this figure was rearranged to Fig. 1g in the revised manuscript.

8) Line 203: the authors probably forgot to delete “as well as”.

Response:

Thanks for your careful reading. We have corrected this mistake in the revised manuscript.

9) Extended figure 4C: what is the time of collection of the tumors?

Response:

Tumors were collected 2 days after YB1 treatment and then applied to immunohistochemical staining of CD3. We have updated time points in the revised manuscript accordingly.

10) Line 342-344: the authors should gain clarity if they precise in the text that the 4T1-Luci cells were injected in day 4 post YB1 treatment, which corresponds to a high IFN- γ serum concentration even in NK cell depleted mice.

Response:

In this experiment, the luciferase labeled 4T1 cells (4T1-Luci) were injected at day 4 post YB1 treatment when the serum IFN- γ concentration was still high. However, in the absence of NK cells, high concentration of IFN- γ alone cannot inhibit formation of lung metastasis. This data indicates that cytokine IFN- γ requires NK cells to exert its anti-metastasis activity.

11) Line 675: Inappropriate capital letter on “Day”

Response: We have corrected this mistake in the revised manuscript.

Reference:

- 1 Delconte, R. B. *et al.* CIS is a potent checkpoint in NK cell-mediated tumor immunity. *Nat Immunol* **17**, 816-824, doi:10.1038/ni.3470 (2016).
- 2 Dong, W. *et al.* The Mechanism of Anti-PD-L1 Antibody Efficacy against PD-L1-Negative Tumors Identifies NK Cells Expressing PD-L1 as a Cytolytic Effector. *Cancer Discov* **9**, 1422-1437, doi:10.1158/2159-8290.CD-18-1259 (2019).
- 3 Nishikado, H., Mukai, K., Kawano, Y., Minegishi, Y. & Karasuyama, H. NK cell-depleting anti-asialo GM1 antibody exhibits a lethal off-target effect on basophils in vivo. *J Immunol* **186**, 5766-5771, doi:10.4049/jimmunol.1100370 (2011).
- 4 Godfrey, D. I. & Berzins, S. P. Control points in NKT-cell development. *Nat Rev Immunol* **7**, 505-518, doi:10.1038/nri2116 (2007).
- 5 Vantourout, P. & Hayday, A. Six-of-the-best: unique contributions of gammadelta T cells to immunology. *Nat Rev Immunol* **13**, 88-100, doi:10.1038/nri3384 (2013).

- 6 Shultz, L. D. *et al.* Multiple defects in innate and adaptive immunologic function in NOD/LtSz-scid mice. *J Immunol* **154**, 180-191 (1995).
- 7 Kandarian, F., Sunga, G. M., Arango-Saenz, D. & Rossetti, M. A Flow Cytometry-Based Cytotoxicity Assay for the Assessment of Human NK Cell Activity. *J Vis Exp*, doi:10.3791/56191 (2017).
- 8 Karimi, M. A. *et al.* Measuring cytotoxicity by bioluminescence imaging outperforms the standard chromium-51 release assay. *PLoS One* **9**, e89357, doi:10.1371/journal.pone.0089357 (2014).
- 9 Jang, Y. Y. *et al.* An improved flow cytometry-based natural killer cytotoxicity assay involving calcein AM staining of effector cells. *Ann Clin Lab Sci* **42**, 42-49 (2012).
- 10 Riccardi, C., Santoni, A., Barlozzari, T., Puccetti, P. & Herberman, R. B. In vivo natural reactivity of mice against tumor cells. *Int J Cancer* **25**, 475-486, doi:10.1002/ijc.2910250409 (1980).
- 11 Kupz, A. *et al.* NLRC4 inflammasomes in dendritic cells regulate noncognate effector function by memory CD8(+) T cells. *Nat Immunol* **13**, 162-169, doi:10.1038/ni.2195 (2012).
- 12 Kupz, A., Curtiss, R., 3rd, Bedoui, S. & Strugnell, R. A. In vivo IFN-gamma secretion by NK cells in response to *Salmonella typhimurium* requires NLRC4 inflammasomes. *PLoS One* **9**, e97418, doi:10.1371/journal.pone.0097418 (2014).
- 13 Li, C. X. *et al.* 'Obligate' anaerobic *Salmonella* strain YB1 suppresses liver tumor growth and metastasis in nude mice. *Oncol Lett* **13**, 177-183, doi:10.3892/ol.2016.5453 (2017).
- 14 Spits, H., Bernink, J. H. & Lanier, L. NK cells and type 1 innate lymphoid cells: partners in host defense. *Nat Immunol* **17**, 758-764, doi:10.1038/ni.3482 (2016).
- 15 Fuchs, A. ILC1s in Tissue Inflammation and Infection. *Front Immunol* **7**, 104, doi:10.3389/fimmu.2016.00104 (2016).
- 16 Eken, A. & Donmez - Altuntas, H. in *Innate Lymphoid Cells (Non - NK ILCs)* (ed Gheorghita Isvoranu) (2017).
- 17 Artis, D. & Spits, H. The biology of innate lymphoid cells. *Nature* **517**, 293-301, doi:10.1038/nature14189 (2015).

REVIEWER COMMENTS

Reviewer #1 (Remarks to the Author):

The authors have made considerable progress against addressing the concerns raised in the primary review.

The only remaining issue I have is with the emphasis on the strain used - from the data presented in the response to my questions, it seems that other Salmonella may well have the capacity to inhibit secondary cancer development including the parent strain that lacks the anoxic-driven *asd* production, which hence may be better adapted to grow within the cancers. This is now clearer in the manuscript around line 139.

The response - Although the parent Salmonella SL7207 strain also showed similar anti-metastatic effects to Salmonella YB1 (supplementary Fig. 1g), the later one is much less toxic against the host and causes almost no side effects (data not shown). Altogether, these findings suggest a possible general mechanism for Salmonella anti-metastatic activity that is dose-dependent, but independent of cancer type and host genetic background.

suggestion - instead of "later one" use YB1 and state that YB1 is less toxic for the hoist evidenced by.....

Because of the (now) more generalised nature of the phenomenon (i.e. perhaps 'all' Salmonella can do this), the argument for using the specific train YB1 needs to be strengthened. It is not sufficient to say "data not shown" - while it is clear from several studies that aromatic mutants (alone) in humans are too reactogenic to be used widely (PMC257661, CVD 906), some data needs to be presented that YB1 has a different pathology profile in mice than say SL7207, even if this is presented as supplementary data. This should include organ counts of the bacteria (spleen and liver may be enough), weight loss, and splenomegaly data from a small cohort at 2 time points, say week 2 and week 4 post-infection.

Reviewer #2 (Remarks to the Author):

The authors took into accounts several remarks of the reviewers and the manuscript was improved. Nevertheless, some comments need to be re-addressed.

Regarding the number of experiments and mice used, the authors specified that each experiment was performed at least twice. The reviewer understands that the pictures of lungs are representative of one experiment but the reviewer does not understand why the associated graphs do not include the data of both experiments. To give a clear example, the authors said in the legend of the figure 1e “n=6 mice per group of a representative experiment of five independent experiments”. Logically, there are 6 lungs PBS and 6 lungs YB1 on the picture 1e which corresponds to a representative replicate. But the associated graph should be a pool of the data from the 5 replicates, meaning it should have more than 6 points per condition. The same problem is at least found for the figures 1e, 1h, 2h, 2i, 3e, 4c, 6c, 6e, 6g, 6h, 7f, 7g, 7h, 7i. Each time, the graph is representative of one experiment instead of being a pool of the replicates. The reviewer wants to have access to all the data and not only to a representative part of them.

The reviewer agrees with the authors that the discrimination of NK cells and non-NK ILC1 is difficult but it is not impossible. Typically, among the Lin-NKp46+ population, NK cells are Eomes+ and non NK ILC1 are Eomes- in almost all tissues. As non-NK ILC1 are high producers of IFN-g and as they are depleted by anti-asialoGM1 antibody, the way the study is conducted, the authors cannot say that the observed effects are due to NK cells and not to non-NK ILC1. The authors seems to be aware of this limitation in their rebuttal but it does not appear in the manuscript. The reviewer advices to perform experiments to discriminate the role of NK cells and non NK ILC1. At least, the authors need to quantify the NK cells and non-NK ILC1 in the lung in PBS and YB1 conditions. They could also perform an IFN-g staining for both populations in these two conditions. If the authors cannot perform these experiments, the authors have, at least, to include the non-NK ILC1 population in all their conclusions.

We thank all the reviewers for their careful reading of our manuscript and their insightful comments and suggestions. A point-by-point response to the reviewers' comments is given below.

Reviewer #1 (Remarks to the Author):

1) The authors have made considerable progress against addressing the concerns raised in the primary review.

The only remaining issue I have is with the emphasis on the strain used - from the data presented in the response to my questions, it seems that other *Salmonella* may well have the capacity to inhibit secondary cancer development including the parent strain that lacks the anoxic-driven asd production, which hence may be better adapted to grow within the cancers. This is now clearer in the manuscript around line 139.

The response - Although the parent *Salmonella* SL7207 strain also showed similar anti-metastatic effects as *Salmonella* YB1 (supplementary Fig. 1g), the latter one is much less toxic against the host and causes almost no side effects (data not shown). Altogether, these findings suggest a possible general mechanism for *Salmonella* anti-metastatic activity that is dose-dependent, but independent of cancer type and host genetic background.

suggestion - instead of "later one" use YB1 and state that YB1 is less toxic for the hoist evidenced by.....

Response:

Thanks for your careful reading. We have revised accordingly in the revised manuscript.

2) Because of the (now) more generalised nature of the phenomenon (i.e. perhaps 'all' *Salmonella* can do this), the argument for using the specific train YB1 needs to be strengthened. It is not sufficient to say "data not shown" - while it is clear from several studies that aromatic mutants (alone) in humans are too reactogenic to be used widely (PMC257661, CVD 906), some data needs to be presented that YB1 has a different pathology profile in mice than say SL7207, even if this is presented as supplementary data. This should include organ counts of the bacteria (spleen and liver may be enough), weight loss, and splenomegaly data from a small cohort at 2 time points, say week 2 and week 4 post-infection.

Response:

We have published a paper in Sci Rep characterizing the profiles of YB1¹. Engineered oxygen-sensitive *Salmonella* YB1 was generated by placing an essential gene under a hypoxia conditioned promoter in SL7207. As a result, the YB1 strain survives only in anaerobic conditions if without the supplement of diaminopimelic acid (DAP) (**Data 1**)¹. When injected into tumor-bearing nude mice, YB1 grew within the tumor, while being rapidly eliminated from normal tissues (**Data 2**)¹.

The specific accumulation of YB1 in tumor tissues was also confirmed in immunocompetent BALB/c mice (**Data 3**). 12 days post *Salmonella* infection, YB1 can be detected only in tumor tissues, while SL7207 and VNP20009 can be detected in most normal tissues and tumor tissues (**Data 3 a**)². Compared to the SL7207 and VNP20009, YB1 strain showed less adverse effects in CT26 colon tumor-bearing mice, indicated by survival curve and body weight changes (**Data 3 b, c**)². No YB1 was detected in the lung, spleen, or liver, while more than 10⁷ CFU/g of YB1 accumulated in tumors from 4T1 tumor-bearing BALB/c mice 20 days post-infection (**Data 3 d**). Restrained by the concentration of oxygen *in vivo*, *Salmonella* YB1 can be quickly eliminated in immunocompetent BALB/c mice without tumor burden, indicated by CFU test in blood, liver, spleen, and lung (**Data 3 e**). The cytotoxicity of YB1 and SL7202 was compared again in immunocompetent mice without tumor burden (**Data 3 f-g**), indicated by spleen size, bodyweight loss, and photos of mice. 8x10⁶ CFU of YB1 or SL7207 were given to BALB/c mice, respectively, and PBS as a control treatment. Compared to PBS treatment and YB1 treatment, SL7207 treated mice showed serious splenomegaly at both time points (**Data 3 f**). Even with the help of ampicillin to control the infection of SL7207, SL7207 treated mice still have significant adverse effects, indicated by body weight changes (**Data 3g**) and daily photo records (**Data 3h**, separately adhered at the end of this reply letter). SL7207 treatment was almost lethal to C57BL/6J mice and the bodyweight change data is shown in **Data 3 i**.

All the above data together indicated that *Salmonella* YB1 could be used as a safe bacterial vector for anti-tumor therapies. We have added reference papers and supplementary data (**supplementary Fig. 1h-i**) in the revised manuscript.

Data 1¹: a-b Growth rate of various strains (10⁴ bacteria/ml) under aerobic or anaerobic conditions in LB broth without DAP (mean±/sd, each time point represents three individual experiments). c-d as in (a, b) but with DAP. DAP was added to prevent cell lysis under aerobic conditions.

Data 2¹: CFU test of YB1, SL7207, and VNP20009 in breast tumor-bearing nude mice. Nude mice with an MBA-MB-231 tumor received vein injections of YB1, SL7207, or

VNP20009. Mice were euthanized at the indicated time points and blood, heart, kidney, liver, lung, lymph node, spleen, and tumor tissues were collected and homogenized. Bacterial accumulation was evaluated by the CFU test. In SL7207 (a) YB1 (b) or VNP20009 (c) treated mice, CFU counts per gram of most normal organs and tumor (red line) are shown over time (mean \pm sd, each time point represents three individual experiments with 2 mice for each experiment). Statistical significance of tumor group vs. all other groups: *. *P<0.05; **P<0.01.

Data3: **a** CT26 colon tumor-bearing BALB/c mice or Nude mice (tumor initial size ~100 mm³) were injected with YB1, VNP20009, or SL7207 (n=5 each; mean ± standard deviation). 12 days after treatment, the mice of each group were euthanized and the bacteria in the organs were counted. **b-c** Evaluation of adverse effects of bacterial infection in CT26 colon tumor-bearing BALB/c mice treated with PBS, YB1, VNP20009, or SL7207. **(b)** Survival rates and **(c)** body weights (n=5 each; mean ± standard deviation) of the mice. ***P<0.001, PBS or YB1 group vs. VNP20009 or SL7207 treatment. PBS, phosphate-buffered saline. **d** YB1 CFU test of different tissues 20 days after YB1 treatment in 4T1 tumor-bearing BALB/c mice. **e** Distribution of *Salmonella* YB1 in blood, liver, lung, and spleen after *i.v.* injection into BALB/c mice (n=5 mice per each group). **f-h** 18 BALB/c mice were divided into three groups and treated with PBS (n=5), 8x10⁶ CFU of YB1(n=5), or 8x10⁶ CFU of SL7207(n=8), respectively. 100ul of 100mg/ml ampicillin was given to each SL7207 treated mouse (SL7207_A+) on days 5, 6, and 7 by *i.p.* injection to control *salmonella* infection. Mice were killed at two timepoints to harvest spleen **(f)**. Daily body weight changes were monitored before day 15 **(g)** and photos of mice were recorded **(h)**, adhered at the end of this reply letter). **i** C57BL/6J mice were divided into three groups and treated with PBS (n=6), YB1(n=6), and SL7207(n=8), respectively. Bodyweight change ratios were shown.

Reviewer #2 (Remarks to the Author):

The authors took into accounts several remarks of the reviewers and the manuscript was improved. Nevertheless, some comments need to be re-addressed.

1) Regarding the number of experiments and mice used, the authors specified that each experiment was performed at least twice. The reviewer understands that the pictures of lungs are representative of one experiment but the reviewer does not understand why the associated graphs do not include the data of both experiments. To give a clear example, the authors said in the legend of the figure 1e “n=6 mice per group of a representative experiment of five independent experiments”. Logically, there are 6 lungs PBS and 6 lungs YB1 on the picture 1e which corresponds to a representative replicate. But the associated graph should be a pool of the data from the 5 replicates, meaning it should have more than 6 points per condition. The same problem is at least found for the figures 1e, 1h, 2h, 2i, 3e, 4c, 6c, 6e, 6g, 6h, 7f, 7g, 7h, 7i. Each time, the graph is representative of one experiment instead of being a pool of the replicates. The reviewer wants to have access to all the data and not only to a representative part of them.

Response:

Thanks for the kind suggestion. We combined repeats for graphs in the revised manuscript for lung metastasis experiments, such as those in figure 1e, 1f, 1h and 6c.

We didn't combine repeats for graphs mainly because of two reasons: 1) for the cancer cell inoculation experiments, although we grow and count cells to prepare a tube of cancer cell suspension for all mice from different groups as consistent as we could, however, different batches of the cells may have differences in their growth condition, and the cell counting itself

may also cause inaccuracy; 2) mice used may differ in their conditions (a combination of 6-8 week-old mice). Therefore, PBS-treated mice from different batches usually develop variable metastatic nodules. For example, we *i.v.* injected 1×10^5 4T1 cells into each BALB/c mouse in Fig. 1e and the representative data was shown in the manuscript. Other independent repeats (**numbered as R-Fig. 1e-ii/iii/iv/v**) are shown below. The tumor nodules of PBS treated mice from different batches were variable, causing bigger error bars if merged to generate graphs. However, every batch of experiment showed the same phenomenon that YB1 treatment potentially blocked the metastasis of 4T1 cancer cells.

As for live imaging, the factors that contribute to the variations from different batches of experiments are more complicated. In addition to the variations derived from cancer cell inoculation and counting of *Salmonella* YB1, the luciferase catalyzed signal is also affected by the substrate stability, the time after cancer cell inoculation, the time between substrate injection and imaging, and the exposure time for imaging. Therefore, the absolute signal intensity is significantly variable for different batches of repeats. What we can guarantee is keeping all mice from each batch of experiment under the same conditions (imaging all groups together). Taking **R-Fig. 2g,h-ii** and **R-Fig. 2g,h-iii** as an example, the conditions for the two batches of repeats are the same, but the signal intensity ranges are variable and cannot be merged to draw a graph even after normalization. We think analyzing live imaging data from different batches of experiments separately is more reasonable and keep figure 2h, 2i, 3e, 4c, 6e unchanged in the manuscript. However, we put all batches of repeated experiments for live imaging as below.

The percentage and absolute cell number of NK cells in the lung are quite stable between different batches of repeated experiments, which enable us to combine repeated experiments to generate graphs for figure 7f, 7g (shown in the revised manuscript). However, flow analysis of NK cells in figure 6g, 6h, 7h, and 7i needs *ex vivo* stimulation of NK cells for several hours, which brings significant variations for different batches of experiments and we can only analyze them separately.

We understand the concerns of reviewers and provide all batches of repeated experimental results as below for all mentioned figures. We name repeated figures with an initial of “R”. For example, R-Fig. 1e-ii/iii/iv/v are repeated experiments for Fig. 1e in the manuscript.

R-Fig. 1e-ii/iii/iv/v Picture of lungs fixed in Bouin solution and quantification (unpaired t-tests) of 4T1 lung metastases after YB1 treatment in the 4T1-BALB/c experimental metastasis model (n=4~5 mice per group). All data are shown as the mean +/- s.e.m.

R-Fig. 1e-ii Picture of lungs and quantification (unpaired t-tests) of lung metastases after YB1 treatment in experimental metastasis model established by inoculation of 2×10^5 B16F10 melanoma cells to each C57BL/6J mouse (n=4 PBS, n=4 YB1). All data are shown as the mean +/- s.e.m. **R-Fig. 1e-iii** Picture of lung metastasis after YB1 treatment in an experimental metastasis model established by inoculation of 8×10^5 or 1.2×10^6 B16F10 melanoma cells to each C57BL/6J mouse (n=2 for each condition).

R-Fig. 2g, h-ii

R-Fig. 2g, h-iii

R-Fig. 2g, h-ii/iii Comparison of live imaging between PBS treated mice and YB1 treated mice has been repeated several times in the original manuscript when we investigated the roles of IFN- γ and NK cells, such as in Fig. 3e, Fig. 6d, Fig. 7d. Here we re-draw graphs using data from Fig. 3e (**R-Fig. 2g, h-ii**) and Fig. 6d, e (**R-Fig. 2g, h-iii**). Tracking and quantification (unpaired t-test) of 4T1-Luci cells *in vivo* by luciferase live imaging 3 h after *i.v.* injection of 4T1-Luci into BALB/c mice. Mice were pretreated with PBS or YB1 3 days before 4T1-Luci cell injection (n=5 mice per group). All data shown as the mean \pm s.e.m.

R-Fig. 2i-ii

R-Fig. 2i-ii Tracking and quantification (unpaired t-test) of MB49-Luci cells *in vivo* by luciferase live imaging 3 h after *i.v.* injection of MB49-Luci into C57BL/6J mice pretreated 3 days in advance with YB1 or PBS (n=3 mice for PBS group, n=5 mice for YB1 group). All data are shown as the mean \pm s.e.m.

R-Fig. 3e-ii

R-Fig. 3e-ii Tracking and quantification (two-tailed unpaired t-tests) of 4T1-Luci cells *in vivo* by luciferase live imaging 3 h after *i.v.* injection of 4T1-Luci into BALB/c mice pretreated with PBS, YB1, or YB1 plus IFN- γ depletion antibody. Mice were pretreated with PBS or YB1 3 days before the injection of 4T1-Luci cells (n=4 mice per group). All data are shown as the mean \pm s.e.m.

R-Fig. 4c-ii

R-Fig. 4c-ii Tracking and quantification (two-tailed unpaired t-test) of 4T1-Luci cells *in vivo* by luciferase live imaging 3 h after *i.v.* injection of 4T1-Luci into NOD SCID mice pretreated 3 days in advance with PBS or YB1 (n=4 mice per group). All data shown as the mean \pm s.e.m.

R-Fig. 6c-ii

R-Fig. 6c-ii Picture and quantification (two-tailed unpaired t-tests) of 4T1 lung metastases across three groups (n=4 mice for YB1-NK depletion group, n=4 for PBS and YB1 groups). 4% PFA fixed tissues are shown. Transparent nodules indicate lung metastasis. All data shown as the mean \pm s.e.m.

R-Fig. 6d, e-ii

The comparison of live imaging between PBS treated mice, YB1 treated mice and YB1 plus NK depletion antibody-treated mice have been done twice and showed in Fig.6d, e and Fig. 7 d, e in the original manuscript. **R-Fig. 6d, e-ii** Tracking and quantification (two-tailed unpaired t-tests) of 4T1-Luci cells *in vivo* by luciferase live imaging 270 minutes after *i.v.* injection of 4T1-Luci into BALB/c mice pretreated with PBS, YB1, or YB1 plus NK depletion antibody. Mice were pretreated with PBS or YB1 4 days before the injection of 4T1-Luci (n=4 mice per group). The data was from Fig. 7d, e of the original manuscript. All data shown as the mean \pm s.e.m.

R-Fig. 6g-ii Flow cytometric analysis of IFN- γ production in lung infiltrating NK cells across samples after co-culture with PMA and Ionomycin *ex vivo* for 5 h (at least 4 biological replicates per group). Both data are shown as the mean \pm s.e.m. *P*-values were derived using a two-tailed unpaired t-test. We didn't combine two batches of results to draw a graph because *ex vivo* stimulations always have considerable variations.

R-Fig. 6h-ii Flow cytometric analysis of CD107a expression on lung infiltrating NK cells across samples after co-culture with YAC-1 cells *ex vivo* for 5 h ($n=4$ biological replicates). Both data are shown as the mean \pm s.e.m. *P*-value was derived using a two-tailed unpaired t-test. We didn't combine two batches of results to draw a graph because *ex vivo* stimulations always have considerable variations.

Fig. 7f**Fig. 7g**
Fig.7 f The percentage of NK cells to all immune cells in lung across samples (n=7 mice per group). **g** The absolute total number of NK cells per lung across samples (n=7 mice per group). Total immune cell numbers were measured by trypan blue exclusion and then multiplied by the percentage of NK cells determined by FACS analysis to give the absolute number of NK cells for each lung. **f-g** Displayed is a combined results of 2 independent experiments. All data are shown as the mean \pm s.e.m. All *p*-values were derived using two-tailed unpaired t-tests. **P*<0.05, ***P*<0.01, ****P*<0.001, *****P*<0.0001.

R-Fig.7h-ii Flow cytometric analysis of IFN- γ production in lung infiltrating NK cells across samples after co-culture with PMA and Ionomycin *ex vivo* for 5.5 h (3 biological replicates per group). Both data are shown as the mean \pm s.e.m. *P*-value was derived using a two-tailed unpaired t-test.

R-Fig.7i-ii Flow cytometric analysis of CD107a expression on lung infiltrating NK cells across samples after co-culture with YAC-1 cells *ex vivo* for 6 h (n=3 mice per group). Both data shown as the mean +/- s.e.m. *P*-value was derived using a two-tailed unpaired t-test.

2) The reviewer agrees with the authors that the discrimination of NK cells and non-NK ILC1 is difficult but it is not impossible. Typically, among the Lin-NKp46+ population, NK cells are Eomes+ and non NK ILC1 are Eomes- in almost all tissues. As non-NK ILC1 are high producers of IFN-g and as they are depleted by anti-asialoGM1 antibody, the way the study is conducted, the authors cannot say that the observed effects are due to NK cells and not to non-NK ILC1. The authors seems to be aware of this limitation in their rebuttal but it does not appear in the manuscript. The reviewer advises to perform experiments to discriminate the role of NK cells and non NK ILC1. At least, the authors need to quantify the NK cells and non-NK ILC1 in the lung in PBS and YB1 conditions. They could also perform an IFN-g staining for both populations in these two conditions. If the authors cannot perform these experiments, the authors have, at least, to include the non-NK ILC1 population in all their conclusions.

Response:

As suggested by the reviewer, we performed flow cytometry analysis and tried to distinguish NK cells and ILC1s by using the Eomes marker. The gating strategy and isotype staining control of Eomes are shown below (**Data 4 a**). We found almost all Lin-NKp46+ cells (more than 96%) are Eomes positive (**Data 4b, c**). After YB1 treatment, only a little bit Lin- NKp46+ cells (~ 4%) showed Eomes negative, which is defined as ILC1s. Besides, the Lin-NKp46+Eomes- cell population didn't show any higher ability to produce IFN- γ after co-culture with cancer cells

(Data 4b). We quantified the ratio of NK cells and ILC1s to all Lin-NKp46+ cells by combining two repeated experiments (**Data 4d**). Together with the results that CD3-NKp46+ cells have significantly higher levels of perforin and granzyme B after YB1 treatment (**Supplementary Fig. 8d-e** in the original manuscript), we still prefer to believe that it is the classical cytotoxic NK cells provoked by YB1 treatment that mediated cancer metastasis suppression. We appreciate the scientific rigor of the reviewers and addressed this concern about ILC1s in the supplementary figure 8 in the revised manuscript.

c

PerCP-eFluor 710
isotype Staining control

d

Data 4 a Gating strategy to distinguish classical NK cells and ILC1s. Gating of Lin- NKp46⁺ was applied to singlets and further divided into two gates: Eomes⁻ cells (ILC1s) and Eomes⁺ cells (NK cells). IFN- γ signal was checked on ILC1s and NK cells. This sample was only stained by anti-Lin antibody cocktail, anti-NKp46 antibody, isotype control of anti-Eomes antibody, and isotype control of anti-IFN- γ antibody. **b** one

representative experiment of the discrimination of ILC1s and NK cells, as well as their IFN- γ level after *ex vivo* co-culture with YAC-1 cells for 4 h. **c** another representative experiment of the discrimination of ILC1s and NK cells. **d** ratios of NK cells and ILC1s to total lung infiltrating Lin-NKp46+ cells. The graph was generated by combining two independent experiments (b-c).

- 1 Yu, B. *et al.* Explicit hypoxia targeting with tumor suppression by creating an "obligate" anaerobic Salmonella Typhimurium strain. *Sci Rep* **2**, 436, doi:10.1038/srep00436 (2012).
- 2 Yu, B. *et al.* Obligate anaerobic Salmonella typhimurium strain YB1 treatment on xenograft tumor in immunocompetent mouse model. *Oncol Lett* **10**, 1069-1074, doi:10.3892/ol.2015.3302 (2015).

Data 3, h

PBS

YB1

SL7207_A+

Day 4

Day 5

Day 6

Day 8

Day 10

(To be continued)

PBS

YB1

SL7207_A+

Day 13

Day 15

Day 18

Day 20

REVIEWERS' COMMENTS

Reviewer #1 (Remarks to the Author):

The authors have made a full response to the queries I raised.

I did NOT question that the YB1 had the DAP phenotype that was claimed, and that a 'DAP-less death' was the likely outcome for any Salmonella strains that are unable to make DAP. The major concern I had was the implication from the writing was that the anti-tumor activity was a YB1-specific phenomenon. The authors acknowledge that it is not and that all Salmonellae (tested) seem to have a predilection for growth in experimental tumors, demonstrated by the data in Data 2. THIS MUST BE CLEAR IN THE RE-WRITING OF THE MANUSCRIPT. While there might be other reasons for choosing YB1, and based on the data shown, the reasons are not efficacy-related, but rather safety-related.

While there some issues with the data as presented in the Figure labelled Data 2 (for example, the time period studied was truncated to 11 days for strains other than YB1), the data demonstrates an attenuation of YB1 that is significantly greater than for the other 2 Salmonella strains, judged by the counts in the tumors, compared with other sites. It is not clear why SL7207, an AroA mutant, is killing BLAB/c mice (Data 3, Panel b) but pre-chorismate mutants cause the splenomegaly seen. And the reduced organ size in YB1 is again clear evidence of attenuation.

To resolve this issue, I would humbly suggest that the authors add a short concluding paragraph to the Discussion that accommodates both the large body of literature relating to the anti-cancer properties of SL7207 (at least 29 papers, many as a vehicle to deliver a DNA vaccine, <https://pubmed.ncbi.nlm.nih.gov/?term=SL7207+cancer&sort=date&size=200>). This paragraph could claim that YB1, is a significant refinement on SL7207, which retains the anti-tumor potency but greatly reduces the reactogenicity of Salmonellae in mice, and that the anti-tumor mechanism appears to be mediated through NK cells releasing IFNg.

Reviewer #3 (Remarks to the Author):

I am satisfied with the m/s now

We thank reviewers for their careful reading of our manuscript and their insightful comments and suggestions. A point-by-point response to the reviewers' comments is given below.

Reviewer #1 (Remarks to the Author):

The authors have made a full response to the queries I raised. I did NOT question that the YB1 had the DAP phenotype that was claimed, and that a 'DAP-less death' was the likely outcome for any *Salmonella* strains that are unable to make DAP. The major concern I had was the implication from the writing was that the anti-tumor activity was a YB1-specific phenomenon. The authors acknowledge that it is not and that all *Salmonellae* (tested) seem to have a predilection for growth in experimental tumors, demonstrated by the data in Data 2. THIS MUST BE CLEAR IN THE RE-WRITING OF THE MANUSCRIPT. While there might be other reasons for choosing YB1, and based on the data shown, the reasons are not efficacy-related, but rather safety-related.

While there some issues with the data as presented in the Figure labelled Data 2 (for example, the time period studied was truncated to 11 days for strains other than YB1), the data demonstrates an attenuation of YB1 that is significantly greater than for the other 2 *Salmonella* strains, judged by the counts in the tumors, compared with other sites. It is not clear why SL7207, an *AroA* mutant, is killing BLAB/c mice (Data 3, Panel b) but pre-chorismate mutants cause the splenomegaly seen. And the reduced organ size in YB1 is again clear evidence of attenuation.

To resolve this issue, I would humbly suggest that the authors add a short concluding paragraph to the Discussion that accommodates both the large body of literature relating to the anti-cancer properties of SL7207 (at least 29 papers, many as a vehicle to deliver a DNA vaccine, <https://pubmed.ncbi.nlm.nih.gov/?term=SL7207+cancer&sort=date&size=200>). This paragraph could claim that YB1, is a significant refinement on SL7207, which retains the anti-tumor potency but greatly reduces the reactogenicity of *Salmonellae* in mice, and that the anti-tumor mechanism appears to be mediated through NK cells releasing IFN γ .

Response: Thanks for your careful reading and critical thinking. In Data 1 as below, 11 days post-treatment, most mice in *Salmonella* SL7207 treated group died and mice in *Salmonella* VPN20009 treated group were too weak to continue the experiment (mice reached the humane endpoint required by the Committee on the Use of Live Animals in Teaching and Research: the body weight loss was more than 20% of the original body weight). As a result, we only had YB1 treated group at a later stage (on day 26). Compared to the other two *Salmonella* strains, only the CFU of YB1 in normal tissues decreased as time went on (Data 1 in the reply letter). This DAP-restricted phenotype of YB1 allows its specific survival in tumor tissue only, thus greatly decreased the toxicity in mice. Therefore, when given at the same dosage (5×10^7 CFU for Data 2a)¹, *Salmonella* SL7207 tends to kill Balb/c mice, while *Salmonella* YB1 won't. Moreover, higher toxicity doesn't necessarily mean stronger anti-metastasis immunity, supported by our result that *Salmonella* SL7207 didn't show stronger anti-metastatic ability than YB1 even though

the SL7207 strain showed much higher toxicity (Dosage: 8×10^6 CFU for each *Salmonella* strains in Data 2b-d). In other publications involving the usage of *Salmonella* SL7207, SL7207 treated mice didn't die because *Salmonella* SL7207 was usually orally or intranasally administered²⁻⁵. If SL7207 was i.v. injected into mice, a lower dosage (5×10^6 CFU) was used and these mice still needed two weeks to recover their body weight⁶.

We are sorry for the inaccurate implication from the writing that the anti-metastasis ability is *Salmonella* YB1 specific, and we have made it clearer in the original manuscript by adding clarifications in the first discussion paragraph, as suggested by the reviewer. Together with the comparison of SL7207 strain to YB1 strain in the result part (**supplementary Fig. 1g-i**), it is quite clear that the anti-metastasis ability is not YB1 strain-specific and YB1 is a significant refinement on SL7207. The clarifications we have added in the manuscript as follow:

Widely used as a DNA vaccine delivery system, *Salmonella* SL7207 showed promising suppressive effect on the growth of primary tumors, especially when carrying all kinds of therapeutic payloads^{3,5}. However, nonnegligible toxicity of *Salmonella* SL7207 on the host dampened its further use for anti-tumor therapies⁶. YB1, a significant refinement on SL7207, retains the same anti-tumor potency but greatly reduces the reactogenicity of *Salmonella* in mice (supplementary Fig. 1g-i). Based on the discovery that *Salmonella* YB1 suppressed metastasis in a liver cancer model⁷, we systemically studied the anti-metastatic ability of *Salmonella* (YB1 as a representative) to unravel the underlying mechanisms.

Data 1: CFU test of YB1, SL7207, and VNP20009 in breast tumor-bearing nude mice. Nude mice with an MBA-MB-231 tumor received vein injections of YB1, SL7207, or VNP20009. Mice were euthanized at the indicated time points and blood, heart, kidney, liver, lung, lymph node, spleen, and tumor tissues were collected and homogenized. Bacterial accumulation was evaluated by the CFU test. In SL7207 (a) YB1 (b) or VNP20009 (c) treated mice, CFU counts per gram of most normal organs and tumor (red line) are shown over time (mean \pm sd, each time point represents three individual experiments with 2 mice for each experiment). Statistical significance of tumor group vs. all other groups: * P<0.05; ** P<0.01.

Data 2: **a** 5×10^7 colony-forming units (CFU) of YB1, VNP20009, or SL7207 were injected intravenously into CT26 tumor-bearing mice through the tail vein. PBS as a control treatment. Survival rates showed the adverse effects of each treatment ($n=5$ for each group). **b-d** 18 BALB/c mice were divided into three groups and treated with PBS ($n=5$), 8×10^6 CFU of YB1 ($n=5$), or 8×10^6 CFU of SL7207 ($n=8$), respectively. 100ul of 100mg/ml ampicillin was given to each SL7207 treated mouse (SL7207_A+) on days 5, 6, and 7 by *i.p.* injection to control *salmonella* infection. Daily body weight changes were monitored before day 15 (**b**). Mice were killed at two time points to harvest the spleen (**c**). For mice sacrificed on day 23, 1×10^5 4T1 cancer cells were *i.v.* injected into each mice on day 6. Lung metastasis of the 10 mice sacrificed on day 23 was quantified (**d**).

Reviewer #3 (Remarks to the Author):

I am satisfied with the m/s now.

Response: Thanks for your careful reading.

1. Yu, B., *et al.* Obligate anaerobic *Salmonella typhimurium* strain YB1 treatment on xenograft tumor in immunocompetent mouse model. *Oncol Lett* **10**, 1069-1074 (2015).
2. Paglia, P., Medina, E., Arioli, I., Guzman, C.A. & Colombo, M.P. Gene transfer in dendritic cells, induced by oral DNA vaccination with *Salmonella typhimurium*, results in protective immunity against a murine fibrosarcoma. *Blood* **92**, 3172-3176 (1998).
3. Mei, Y., *et al.* Combining DNA Vaccine and AIDA-1 in Attenuated *Salmonella* Activates Tumor-Specific CD4(+) and CD8(+) T-cell Responses. *Cancer Immunol Res* **5**, 503-514 (2017).
4. Berger, E., *et al.* *Salmonella* SL7207 application is the most effective DNA vaccine delivery method for successful tumor eradication in a murine model for neuroblastoma. *Cancer Lett* **331**, 167-173 (2013).

5. Cao, H., *et al.* MDA7 combined with targeted attenuated Salmonella vector SL7207/pBud-VP3 inhibited growth of gastric cancer cells. *Biomed Pharmacother* **83**, 809-815 (2016).
6. Felgner, S., *et al.* Engineered Salmonella enterica serovar Typhimurium overcomes limitations of anti-bacterial immunity in bacteria-mediated tumor therapy. *Oncoimmunology* **7**, e1382791 (2018).
7. Li, C.X., *et al.* 'Obligate' anaerobic Salmonella strain YB1 suppresses liver tumor growth and metastasis in nude mice. *Oncol Lett* **13**, 177-183 (2017).